# Development of *Plasmodium falciparum* liver-stages in hepatocytes derived from human fetal liver organoid cultures

Annie S. P. Yang [1,14] ✉, Devanjali Dutta[2,3,8,14], Kai Kretzschmar [2,3,4,14], Delilah Hendriks [2,3,14], Jens Puschhof [2,3,9], Huili Hu [2,3,10], Kim E. Boonekamp [2,3,11], Youri van Waardenburg[1], Susana M. Chuva de Sousa Lopes [5], Geert-Jan van Gemert[1], Johannes H. W. de Wilt [6], Teun Bousema [1], Hans Clevers [2,3,7,12,15] ✉ & Robert W. Sauerwein[1,13,15] ✉

*Plasmodium falciparum* (*Pf*) parasite development in liver represents the initial step of the life-cycle in the human host after a *Pf*-infected mosquito bite. While an attractive stage for life-cycle interruption, understanding of parasite-hepatocyte interaction is inadequate due to limitations of existing in vitro models. We explore the suitability of hepatocyte organoids (HepOrgs) for *Pf*-development and show that these cells permitted parasite invasion, differentiation and maturation of different *Pf* strains. Single-cell messenger RNA sequencing (scRNAseq) of *Pf*-infected HepOrg cells has identified 80 *Pf*-transcripts upregulated on day 5 post-infection. Transcriptional profile changes are found involving distinct metabolic pathways in hepatocytes with Scavenger Receptor B1 (SR-B1) transcripts highly upregulated. A novel functional involvement in schizont maturation is confirmed in fresh primary hepatocytes. Thus, HepOrgs provide a strong foundation for a versatile in vitro model for *Pf* liver-stages accommodating basic biological studies and accelerated clinical development of novel tools for malaria control.

Malaria is one of the major infectious diseases worldwide, responsible for a large and increasing disease burden particularly in sub-Saharan Africa with 247 million cases and 619,000 deaths in 2021[1]. While several *Plasmodium* species can affect humans, the majority of disease burden is caused by *Plasmodium falciparum* (*Pf*). Malaria is contracted by a small number of liver infective *Pf* parasites (sporozoites) injected into the skin of the human host by an infected female mosquito while taking a blood meal. Within maximally a few hours, a number of these

[1]Radboud Center of Infectious Diseases, Department of Medical Microbiology, Radboud University Medical Center, Nijmegen, the Netherlands. [2]Hubrecht Institute, Royal Netherlands Academy of Arts and Sciences, Utrecht, The Netherlands. [3]Oncode Institute, Utrecht, The Netherlands. [4]Mildred Scheel Early Career Centre (MSNZ) for Cancer Research Würzburg, University Hospital Würzburg, Würzburg, Germany. [5]Anatomy and Embryology, Leiden University Medical Center, Leiden, the Netherlands. [6]Department of Surgery, Radboud University Medical Center, Nijmegen, the Netherlands. [7]Princess Maxima Center (PMC) for Pediatric Oncology, Utrecht, the Netherlands. [8]Present address: Merus, Utrecht, the Netherlands. [9]Present address: Microbiome and Cancer Devision, German Cancer Research Center (DKFZ), Im Neuenheimer Feld 280, 69120 Heidelberg, Germany. [10]Present address: The Research Center of Stem Cell and Regenerative Medicine, Department of Systems Biomedicine, School of Basic Medical Sciences, Cheeloo College of Medicine, Shandong University, Jinan, China. [11]Present address: Signaling and Functional Genomics Devision, German Cancer Research Center (DKFZ), Im Neuenheimer Feld 280, 69120 Heidelberg, Germany. [12]Present address: Pharma, Research and Early Development (pRED) of F. Hoffmann-La Roche Ltd, Basel, Switzerland. [13]Present address: TropIQ Health Sciences, Nijmegen, the Netherlands. [14]These authors contributed equally: Annie S. P. Yang, Devanjali Dutta, Kai Kretzschmar, Delilah Hendriks. [15]These authors jointly supervised this work: Hans Clevers, Robert W. Sauerwein. ✉e-mail: annie.yang@radboudumc.nl; h.clevers@hubrecht.eu; r.sauerwein@tropiq.nl

sporozoites reach the liver via the bloodstream to invade hepatocytes. After a clinically silent seven-day period of intracellular replication and differentiation, each sporozoite can give rise to an estimated 10,000 blood-infective daughter parasites (merozoites). These merozoites are released from ruptured hepatocytes into the bloodstream where they start their 48-h cycle of asexual replication in circulating red blood cells, responsible for clinical disease.

Given the low numbers of sporozoites injected, this early phase of human host infection represents a relative vulnerable and critical stage in the parasite life-cycle before large parasites loads are generated during blood-stage malaria. Therefore, availability of highly effective drugs and/or vaccines targeting these parasite-liver stages would be a great asset for malaria control and elimination. A major contributing factor is the lack of understanding of the basic biology of the dynamic interaction between developing parasites in invaded hepatocytes. This can be best approached by using a non-biased survey of host and parasite transcriptome/proteome during an infection process. A main stumbling block is the absence of representative in vitro liver cell cultures encompassing full *Pf*-development. Cryopreserved primary human hepatocytes have been used but suffer from variable and low permissibility providing insufficient material for such analysis[2]. Alternatively, access and availability of freshly isolated primary human hepatocytes are limited. Hepatocytes from both sources can only be maintained in culture for a limited number of days. Finally, the use of hepatic cell lines including HC-04 is mostly restricted to examination of the earliest events of parasite-hepatocyte interaction (i.e., traversal and invasion), although full development has been reported[3].

Organoids are three-dimensional structures that are derived from stem cells or primary tissues and recapitulate key architectural and functional aspects of the organ/tissue under investigation. When derived from adult stem cells, these in vitro structures can be expanded over long time periods[4]. We have recently established a protocol to grow organoids from human hepatocytes[5]. These organoids can be established from single hepatocytes and grown for multiple months, while retaining key morphological, functional and gene-expression features. Transcriptional profiles of the organoids resemble those of proliferating hepatocytes after partial hepatectomy. Human hepatocyte organoids (HepOrgs) proliferate extensively after engraftment into mice. We have observed an inverse correlation between age of the donor and the length of the exponential expansion phase. In fact, fetal hepatocytes yield organoid lines that can be expanded indefinitely, while still closely resembling primary hepatocytes (HuHeps)[6].

Here, we show *Pf*-permissiveness of a number of differentiated HepOrgs. Furthermore, single-cell messenger RNA-seq has been performed on both uninfected and infected HepOrgs to study profiles of up- and downregulated gene transcripts of both *Pf* and host cells containing mature schizonts. Transcripts of Scavenger Receptor B1 (encoded by gene *SCARB1*; protein referred to as SR-B1) are upregulated and a novel critical function of this liver protein for *Pf*-development is subsequently confirmed in fresh primary HuHeps.

## Results

### Complete development of PfNF175 liver stages in HepOrg

Four different fetal HepOrg lines (KIFM, KK2, KK3 and KU1) were infected with PfNF175 sporozoites[7,8]. Infection rates were determined at day 5–8 post infection (p.i.) by determining the number of PfHSP70 positive schizonts (Fig. 1A). An infection rate between 0.5–1% was obtained in all four HepOrg lines which is lower than in freshly isolated primary HuHeps[7].

The sizes of HepOrg schizonts were not significantly different from those in primary HuHeps in two independent experiments (HepOrg A and HepOrg B) on days 3, 5 and 7 p.i. (Fig. 1B). Furthermore, schizonts showed similar expression of typical *Pf*-liver-stage markers including circumsporozoite protein (PfCSP), exported protein 2 (PfEXP2), and glyceraldehyde-3-phosphate dehydrogenase (PfGAPDH)

(Fig. 1C, D; Table 1). Finally, HepOrg schizonts showed typical merozoite surface protein 1 (PfMSP1) on day 7 and 8 p.i. which is indicative of maturation (Fig. 1F).

We next examined the sensitivity of NF175 HepOrg schizonts to atovaquone which is an established drug with anti-liver-stage activity[2] (Fig. S1). Treatment with atovaquone (10 nM and 50 nM) reduced infection rate by more than 75%, which is similar to results in HuHeps[2,9]. The remaining schizonts also showed a 90% reduction in size. NF175 was not sensitive to pyrimethamine in both HepOrgs and cryopreserved HuHeps. The combined data show that *Pf*-sporozoites infect and develop in HepOrgs while staying similarly responsive to the established liver-schizonticidal drug atovaquone.

### Transcriptional changes in HepOrgs after Pf infection

We next studied host cell transcripts in NF175-infected HepOrgs and used fluorescence activated cell sorting (FACS) to obtain single cells (both infected and uninfected) (Fig. S2). Isolated primary hepatocytes quickly dedifferentiated in a 2D format in a process known as epithelial-mesenchymal transition[10]. This also occurred in differentiated HepOrgs as shown by a reduction in albumin levels, when cultured in a medium conducive to schizont growth (Fig. S2E). For evaluation of transcriptomic changes in hepatocytes during intracellular parasite development, we tested a number of media with the aim to maintain the hepatic phenotype (readout of albumin) while allowing for parasite development; a mixture of William's B (WLB) and HuHeps liver medium (HLM) used at 1:1 ratio was selected (Fig. S2C–E). In total, 1920 cells were isolated 4 and 5 days p.i. and processed by sorting and robot-assisted transcriptome sequencing (SORT-seq)[11]. Sequences were mapped to reference genomes of both human and *Pf* origin and analyzed using Seurat v4[12]. Single-cell gene-expression profiles of the human reads (n = 1277 cells) grouped into 5 main clusters (Fig. 2A, Fig. S3A–C, Supplementary Data 1, 5 and 6). Clusters 0, 1, 2 and 3 robustly expressed *ALB* and *AFP*, two markers of differentiated hepatocytes (Fig. 2A, B). Cluster 2 was enriched in *MKI67*-expressing proliferative hepatocytes (Fig. 2A, B). Cells in cluster 3 showed the highest transcript counts of *ALB* and *AFP* and other hepatocyte differentiation marker suggesting enrichment for the most differentiated hepatocytes (Fig. 2A–C, Fig. S4). Cluster 4 contained cells expressing high levels of cholangiocyte lineage markers *EPCAM* and *KRT7* (Fig. 2A, B, Fig. S5). Ratios of the main cell clusters were consistent between test-samples (cells exposed to *Pf* parasites) and non-infected controls (Fig. S3D, E).

CD81 is a well-established host-cell ligand for *Pf* hepatocyte entry[13–15] while a similar although controversial role has been proposed for SR-B1 (encoded by the gene *SCARB1*) and EphA2[16–18]. Both *CD81* and *EPHA2* expression were lowest in cluster 3 (Fig. S6A, B), while *SCARB1* (encoding SR-B1) expression was highest in this cluster (Fig. 2H, I). Cluster 3 had the highest number of infected cells (Fig. 2D–F) and, in line, *SCARB1* expression was significantly increased in highly infected cells (i.e., cells with high parasite transcript counts) compared to uninfected cells or cells with low parasite transcript counts (Fig. 2J). In contrast, the lowest transcript counts for *CD81* and *EPHA2* were found in highly infected cells (Fig. S6A, B). In addition, *SCARB1* expression was strongly correlated with hepatocyte markers *ALB*, *AFP* and *RBP4* (r - 0.5) while moderately negatively correlated with the cholangiocyte markers *EPCAM*, *KRT19* and *KRT8* (r < −0.2) (Fig. S6C). The combined data show that *Pf*-parasites do develop in all five identified cell clusters with a strong preference for mature hepatocytes as represented by cluster 3.

In search for potential host-cell differences between infected and uninfected HepOrgs, we sub-clustered the dataset of highly infected cells (n = 348 cells) on day 5 p.i., (Fig. 2G, A). Genes were differentially expressed (DE) in infected- compared to uninfected hepatocytes showing 674 upregulated and 1218 downregulated transcripts respectively in infected cells (Fig. 3B, Supplementary Data 2). Genes associated with lipid metabolism including *APOA1, APOB, FASN, MTTP*

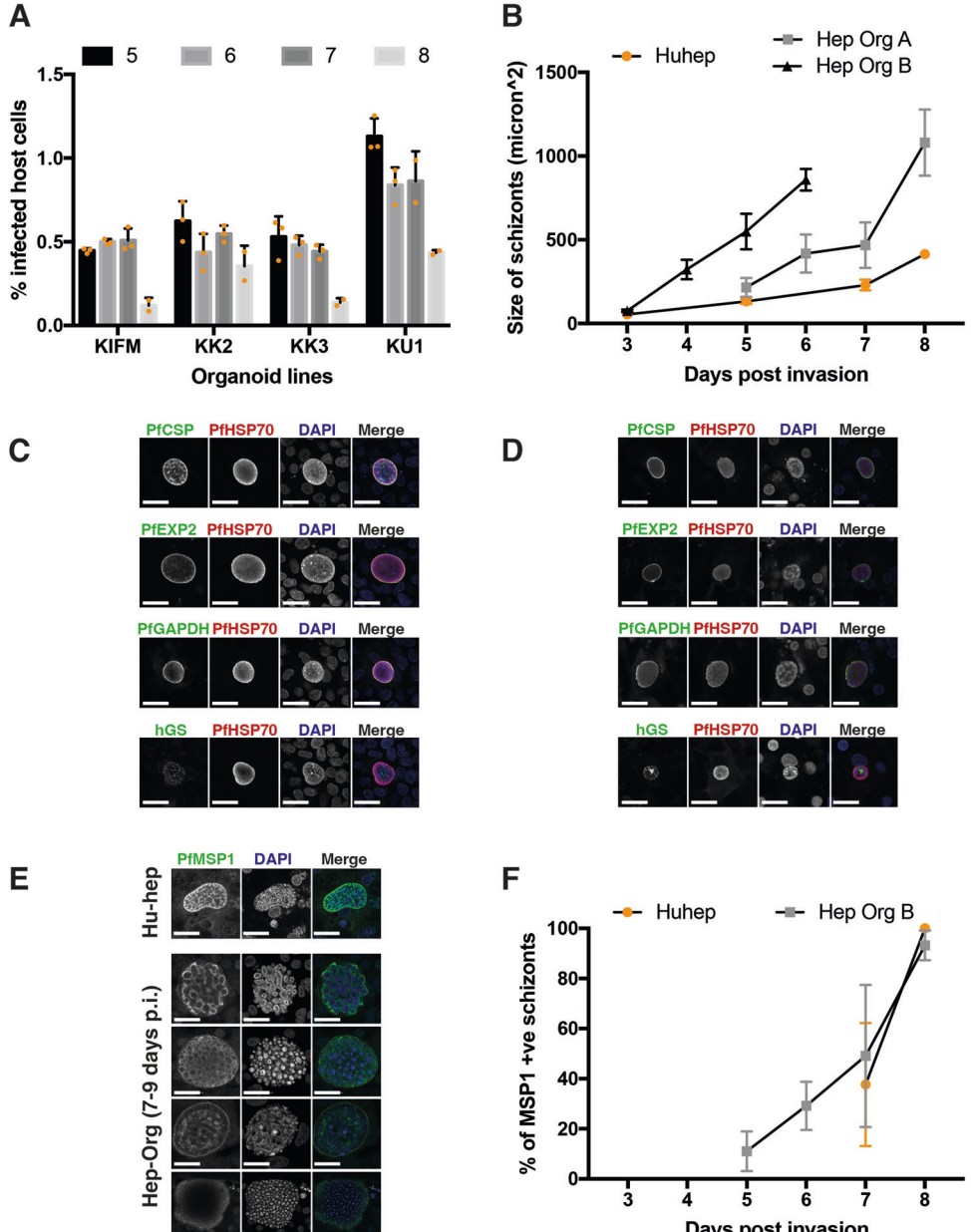

**Fig. 1 | *P. falciparum* NF175 infection and development in HepOrgs. A** Percentage of host cells (HepOrgs) with NF175 schizonts. Each bar represents the average of triplicates while only duplicates are shown for day 8. (*n* = 4 donors; KIFM, KK2, KK3 and KU1; error bar is the mean ± SD of three technical replicates). **B** The size of NF175 schizonts in HepOrg experiment A and B compared to HuHeps. Each point represents the average of median of >100 schizonts from three biological replicates with standard deviation. For the HepOrg A, each point represents the average of median of KK2, KIFM, KU1 where >25 schizonts are measured for each line at each time point. For HepOrg B, each point represents the average of median of KIFM, KK2, KK3 and KU1 where > 100 schizonts are measured for each line at each time point. Areas under the schizont growth curves of Huheps and respectively HepOrg A (*p* = 0.001; *n* = 4 donors) and HepOrg B (*p* = 0.0002; *n* = 4 donors) were significantly different (Welch *t*-test, two-sided). **C** Confocal images of HepOrgs and **D** HuHeps at day 5 p.i. showing the expression of typical liver-stage markers; from top to bottom: Circumsporozoite protein (CSP), Exported Protein 2 (EXP2), Glyceraldehyde-3-phosphate dehydrogenase (GAPDH) and human Glutamine Synthetase (hGS). Scale bar is 25 microns. **E** Confocal images of Merozoite Surface Protein 1 (PfMSP1) in schizonts of HuHeps (top) and HepOrgs (bottom) on day 7-9 p.i. Scale bar is 25 microns. For **C**, **D** and **E**, these stainings have been repeated in three independent experiments and here a representative image is shown for each staining combination. **F** Average percentage with standard deviation of PfMSP1 positive schizonts from HepOrg B (*n* = 4; KIFM, KK2, KK3, KU1) and HuHep (*n* = 3) assessed >100 schizonts per line per time point. See also Fig. S1.

and *PPARA* as well as genes related to glycolysis such as *G6PC* were significantly upregulated in infected hepatocytes (Fig. 3C, Supplementary Data 2). Unbiased analysis using EnrichR[19] revealed that genes associated with cholesterol metabolism (WP4718), the PPAR signaling pathway (WP3942), nonalcoholic fatty liver disease (WP4396), the metabolic pathway of LDL, HDL and TG, including related diseases (WP4522), fatty acid biosynthesis (WP357), glycolysis and gluconeogenesis (WP534), nuclear receptors in lipid metabolism and toxicity (WP299) and fatty acid beta-oxidation (WP143) were significantly upregulated in infected hepatocytes (Supplementary Data 3). The combined data indicate that *Pf* infection induces host changes in metabolism that accommodate energy consumption.

Finally, the *Pf* transcriptome in infected HepOrgs on day 5 p.i. showed significant upregulation of more than eighty markers including liver-stage-specific *CSP*, *LISP1* and *SLARP* (Fig. 4A, C, D and, Supplementary Data 4). The expression profile was also different from that

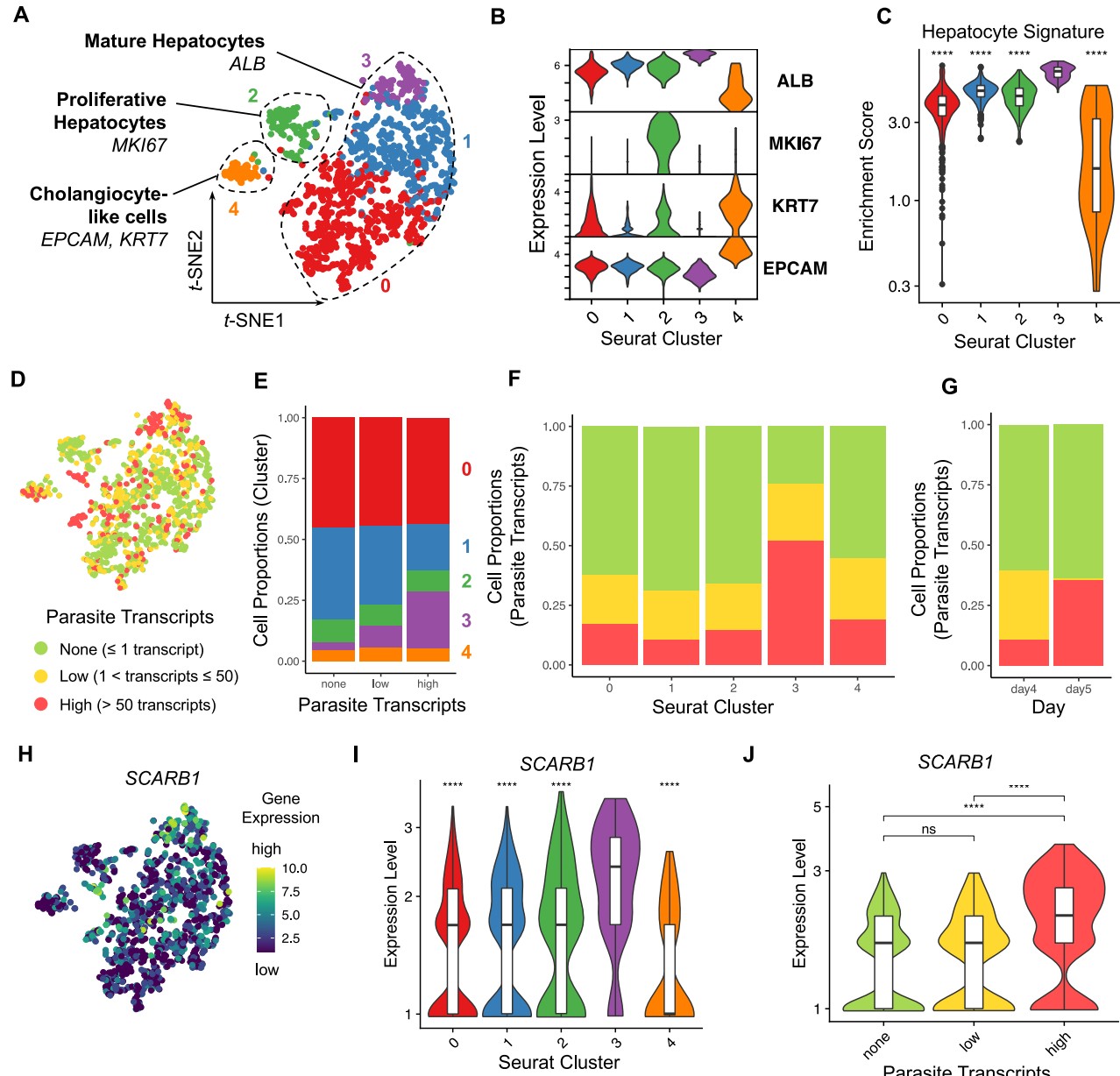

**Fig. 2 | Expression of hepatocyte maturation markers in Pf-infected HepOrgs at 4 or 5 days p.i. A** *t*-SNE map showing the Seurat clusters and major cell types in the human read dataset (*n* = 1277 cells; cluster 0, *n* = 571 cells; cluster 1, *n* = 426 cells; cluster 2, *n* = 117 cells; cluster 3, *n* = 100 cells; cluster 4, *n* = 63 cells). **B** Violin plot showing the lineage marker gene expression per Seurat cluster. **C** Violin plot showing the enrichment scores of the hepatocyte gene signature per Seurat cluster. **D** *t*-SNE map highlighting the infection rates of the cells (none, *n* = 786 cells; low, *n* = 269 cells; high, *n* = 222 cells). **E** Column chart showing the cell proportions per Seurat cluster and infection rate. Column chart showing the cell proportions per infection rate and **F** Seurat cluster or **G** collection day (4 or 5 days p.i.). **H** *t*-SNE map

showing *SCARB1* expression. Violin plots showing *SCARB1* expression per **I** Seurat cluster and **J** infection rate. Two-sided Mann–Whitney U (Wilcoxon rank-sum) test: ns $p \geq 0.05$, ****$p < 0.0001$. Statistics in **C** and **I** were calculated in comparison to cluster 3. **C** $p < 2^{-16}$ (all comparisons); **I** $p < 2^{-16}$ (vs cluster 0), $p = 3.1^{-13}$ (vs cluster 1), $p = 8.6^{-8}$ (vs cluster 2), $p = 2.7^{-12}$ (vs cluster 4); **J** $p = 0.53$ (none vs low), $p < 2.22^{-16}$ (none vs high), $p = 1.9^{-14}$ (low vs high). Box plots in **C**, **I** and **J** indicate the median (Q2), 25th percentile (Q1) and 75th percentile (Q3) with the whiskers showing the minimum (Q1 − 1.5 × interquartile range) and maximum (Q3+ 1.5 × interquartile range). See also Fig. S2–S6, S11 and Supplementary Data 1. Source data are provided as a Source data file and Supplementary Data 5.

generated from blood-stage parasites (Fig. 4B, C, Supplementary Data 4, Supplementary Data 7).

**Role of SR-B1 in Pf- liver-stage development in primary HuHeps**
SR-B1 is a well-known host factor involved in liver cell invasion *Plasmodium* sporozoites[16,17]. As shown in Fig. 2I, J, there was a distinct upregulation of SR-B1 transcripts from day 5 p.i., suggestive for a late and unknown functional role in parasite development as reflected by an increased SR-B1 staining in infected HuHeps on day 5 p.i. (Fig. S12). We tested the possible toxic effects of two known SR-BI inhibitors,

Block Lipid Transport BLT1[16,17] and ITX5061[20] in HepOrg and HuHep cultures (Fig S7A). Only ITX5061 showed toxicity in both HepOrg and HuHeps at the two highest concentrations tested. Figure S7C−E showed that a 50% reduction in schizont numbers in HuHeps treated with BLT1 at 20−40 μM but not for ITX5061 which required a higher concentration (40−80 μM). HuHeps, as more directly biologically relevant cells, were treated with BLT1 (20 μM) at different time points (Fig. S8) to examine its impact on parasite development of PfNF54, PfNF135 and PfNF175 on day 7 p.i. (Fig. 5, Fig. S9)[16,21]. When applied 24-h prior to infection (Day −1), BLT1 did not prevent sporozoite invasion of

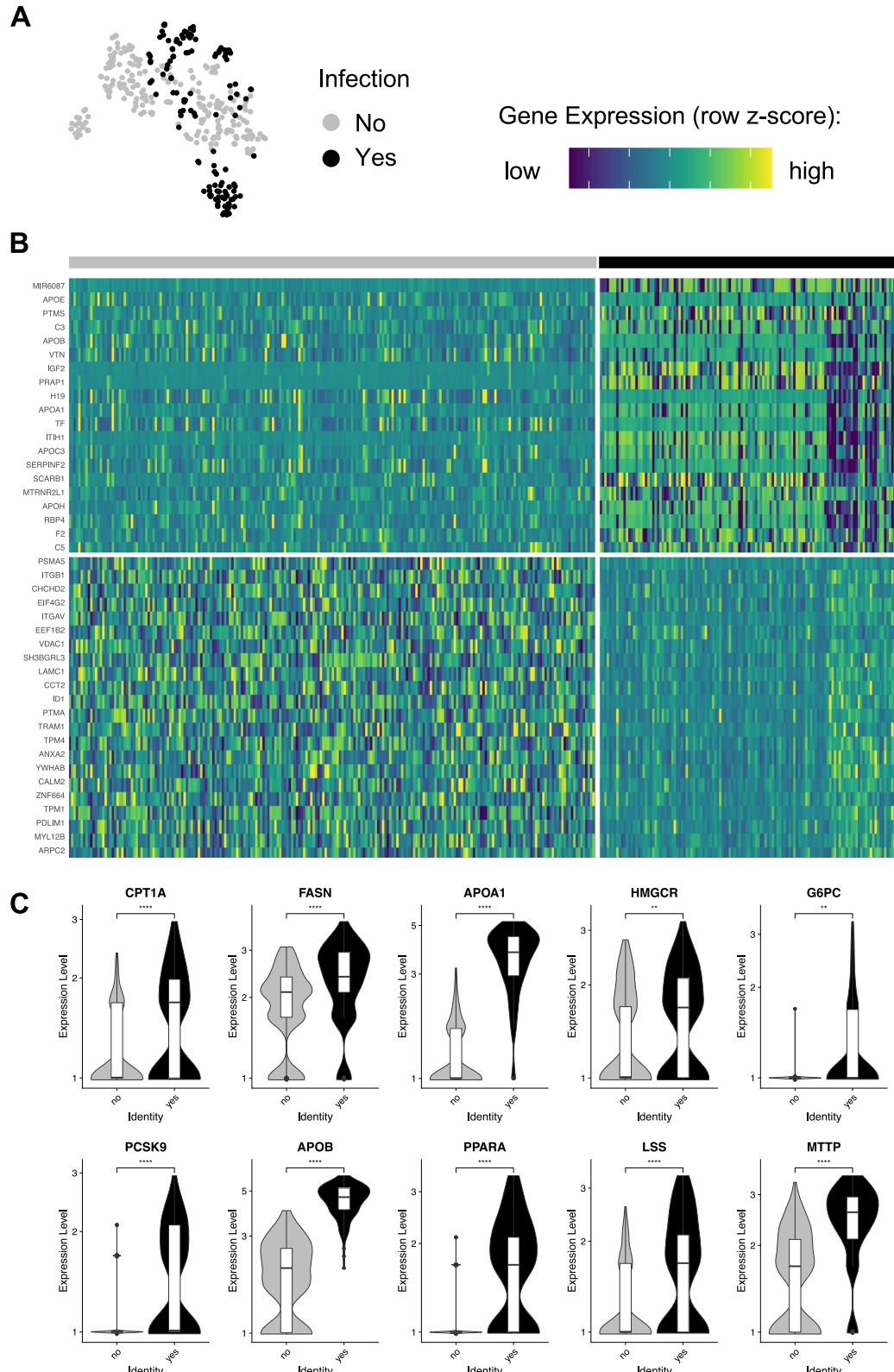

**Fig. 3 | Pf-infected hepatocytes display a distinct gene-expression pattern. A** *t*-SNE map of cells at day 5 p.i. showing the infection state (uninfected cells, *n* = 222; infected cells, *n* = 126). **B** Heatmap showing differentially expressed (DE) genes between infected and uninfected cells. **C** Violin plots showing a selection of DE genes between infected and uninfected cells. Two-sided Mann–Whitney *U* (Wilcoxon rank-sum) test: \*\**p* < 0.01, \*\*\**p* < 0.001, \*\*\*\**p* < 0.0001. *CPT1A*, p = 3.3$^{-5}$; *FASN*, p = 5.8$^{-10}$; *APOA1*, *p* > 2.22$^{-16}$; *HMGCR*, *p* = 0.0062; *G6PC*, *p* = 0.0011; *PCSK9*, *p* = 2$^{-10}$; *APOB*, *p* > 2.22$^{-16}$; *PPARA*, *p* > 2.22$^{-16}$; *LSS*, *p* = 1.3$^{-7}$; *MTTP*, *p* > 2.22$^{-16}$. Box plots in **C** indicate the median (Q2), 25th percentile (Q1) and 75th percentile (Q3) with the whiskers showing the minimum (Q1 − 1.5 × interquartile range) and maximum (Q3 + 1.5 × interquartile range). See also Supplementary Data 2 and 3. Source data are provided as a Source data file and Supplementary Data 5.

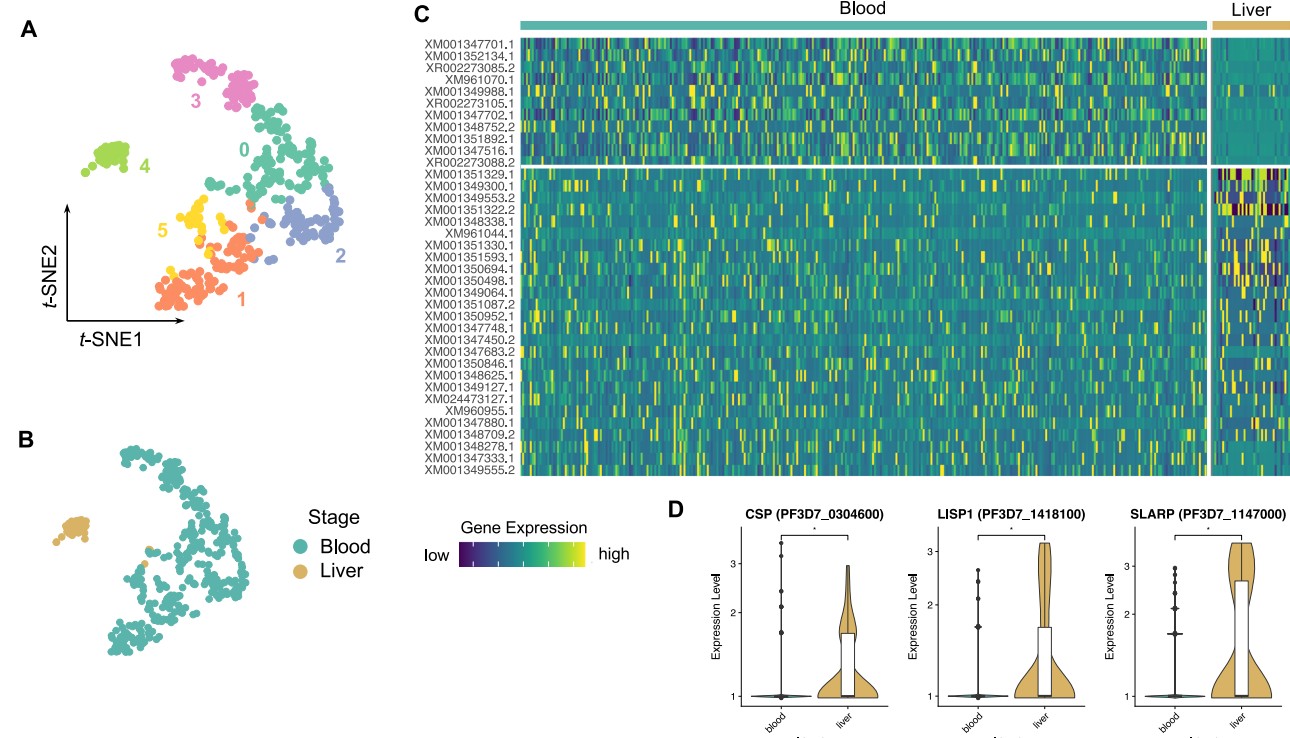

**Fig. 4 | Single-cell transcriptomics shows *Pf*-liver stage-specific gene-expression pattern. A, B** Parasite-derived *t*-SNE maps of individual cells showing Seurat cell clusters (**A**; *n* = 395 cells; cluster 0, *n* = 109 cells; cluster 1, *n* = 90 cells; cluster 2, *n* = 66 cells; cluster 3, *n* = 63 cells; cluster 4, *n* = 36 cells; cluster 5, *n* = 31 cells) and infected HepOrg cells (*Pf* liver stage; *n* = 40 cells) or erythrocytes (*Pf* blood stage; *n* = 355 cells), respectively **B**. Cluster 4 is enriched in infected cells from HepOrgs, while the remaining clusters mostly contain red blood cells. **C** Heatmap showing differentially expressed (DE) genes between liver and blood stage. **D** Violin plots showing DE genes enriched in the liver stage compared to the blood stage. Two-sided Mann–Whitney *U* (Wilcoxon rank-sum) test: *$p < 0.05$. *CSP*, $p = 0.019$; *LISP1*, $p = 0.031$; *SLARP*, $p = 0.031$. Box plots in **D** indicate the median (Q2), 25th percentile (Q1) and 75th percentile (Q3) with the whiskers showing the minimum (Q1 − 1.5 × interquartile range) and maximum (Q3 + 1.5 × interquartile range). See also Supplementary Data 4. Source data are provided as a Source data file and Supplementary Data 7.

NF135 and NF175, in contrast to NF54 (Fig. 5A–C). This indicated that the former strains appear to have alternative SR-B1-independent entry pathways. However, when the infected monolayer was treated from day 1 p.i. onwards, there was a drastic reduction in schizont number for all three *Pf* strains in particular at day 3 and 4 p.i. (Fig. 5A–C). This was also reflected in the reduced schizont size for all *Pf* strains (Fig. 5D–F, Fig. S10). The combined data show that SR-B1 is functionally involved in late-stage liver maturation of all three *Pf* strains tested, while only critical for invasion of NF54.

It has previously been shown that *P. yoelii* and *P. berghei* (*Plasmodium* species of murine malaria models) must scavenge host lipids for successful liver-stage development[22,23]. For their intra-hepatic growth and development, growing parasites need to expand their plasma membranes as well as the surrounding parasitophorous vacuole membrane (PVM). We therefore hypothesized that inhibition of SR-B1 function may interfere with daughter parasite packaging which requires essential lipids. SR-B1 binds to high-density lipoprotein (HDL) and mediates the entry of the cholesteryl ester or other lipids present at the HDL particle core into the hepatocytes[24,25]. BLT1-treated PfNF54-, PfNF135- and PfNF175-infected HuHeps showed a reduced trend of MSP-1 positive liver schizonts on day 3 and 4 (Fig. 6A–C). The remaining MSP-1 positive schizonts showed atypical MSP-1 staining pattern (Fig. 6D–F). The combined data suggest that SR-B1 may be involved in the maturation of liver schizonts.

## Discussion

In this study, we show that differentiated fetal human hepatocyte organoids are amenable to infection with *Pf* parasites and do sustain the mature schizont liver stage. Differentiated hepatocytes generated from fetal HepOrgs provide an alternative in vitro culture system that may combine advantageous characteristics of both primary hepatocytes and hepatic cell lines. Infected HepOrgs have characteristics including morphology and expressed marker profile that are similar to those of primary HuHeps. Schizonts formed in these HepOrgs are generally bigger than those present in HuHeps and closer in size to those observed in mice with humanized livers[26,27]. Furthermore, the presence of MSP-1 expression on late liver stages confirms proper maturation. Importantly, the developing intracellular parasites are sensitive to the well-established schizonticidal drug atovaquone. Overall, our results show that differentiated hepatocytes generated from fetal HepOrgs provide a functional in vitro culture system that meets important limitations of both primary HuHeps and hepatic cell lines. For instance fresh- and/or cryopreserved HuHeps suffer from heterogeneity in *Pf*-permissiveness as well as a limited in vitro cellular lifespan starting to deteriorate already a few days after seeding[28], while hepatic cell lines show only limited support to *Pf*-development[28].

Previously, we have shown a strong correlation between schizont size and intra-parasitic human glutamine synthetase (GS) levels in NF175 HuHeps schizonts[7]. However, ability to accumulate GS does not seem to be the determining factor for schizont size in HepOrgs with GS positive and negative schizonts displaying similar sizes. A possible explanation may relate to GS availability and redistribution within the parasite: HepOrg schizonts show more extensive GS networks compared to the one single concentrated spot present in HuHeps schizonts. A reason may be that HepOrgs are cultured in the presence of

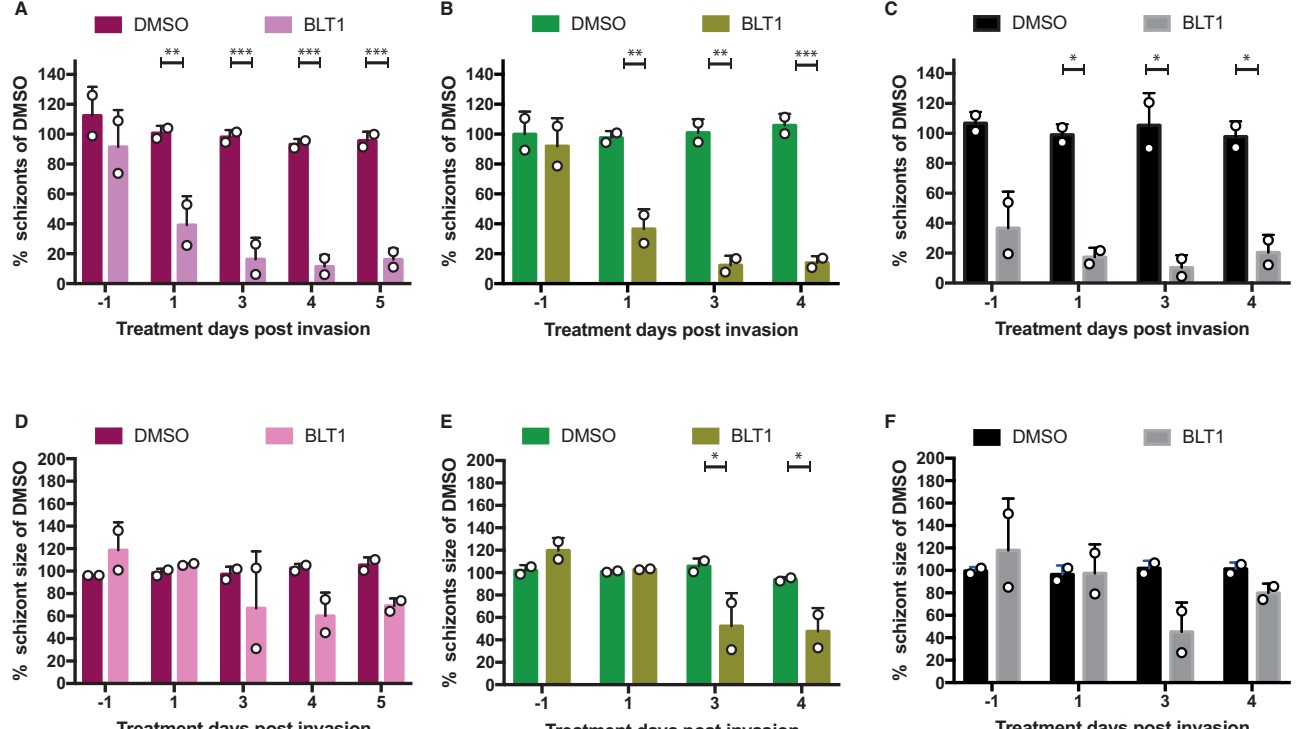

**Fig. 5 | Effect of BLT1-inhibitor on Pf-liver-stage development.** The average (of triplicates) number (**A**–**C**) and size (**D**–**F**) of schizonts in HuHeps infected with strain NF175 (Magenta) NF135 (Green) and NF54 (Black) and treated with either DMSO or BLT1. Each bar represents the average and error bars (SD) of three technical replicates (**A**–**C**) from two biological experiments (*n* = 2). For schizont size (**D**–**F**) data are shown of two technical replicates of at least 100 schizonts except for NF54 due to low numbers of schizonts surviving (**D**–**F**). Raw numbers for this panel are shown in Figs. S8 and S9. Sidak's multiple comparison test (two sided) is used to compare DMSO and BLT1 conditions for **A**–**F** and annotated as *$p < 0.05$, **$p < 0.01$, ***$p < 0.001$. For **A**, $p$ values from left to right are 0.0014, 0.0004, 0.0004, and 0.0005. For **B**, $p$ values are 0.0032, 0.0010, 0.0009. For **C**, $p$ values are 0.0369, 0.0216, 0.0447. For **E**, $p$ values are 0.0182 and 0.0387.

CHIR99021, a strong inhibitor of glycogen synthase kinase 3 (GSK3), a protein that phosphorylates β-catenin targeting proteasomal degradation[29,30]. One of the functions of β-catenin is that of a key Wnt signaling pathway effector, inducing the transcription of various Wnt/β-catenin target genes including *GLUL* (encoding GS)[31]. Therefore, one may assume that HepOrg cells contain an excess of GS (Fig. S11), which can benefit parasite growth even if not transported into the parasite.

Organoid technology emerges as an in vitro model system to overcome some of the challenges of conventional models[32,33]. One advantage of organoids is the potential of forming a three-dimensional cellular cluster that anatomically and functionally better resembles the host organ under study, while also keeping the capacity of self-renewal and self-organization. In case of liver studies, hepatocytes in a two-dimensional monolayer quickly lose their hepatocyte functions[20]. Chua and colleagues introduced a liver spheroids model for malaria liver-stage research based on primary hepatocytes grown in 3D Cel-lusponge to form spheroids[21]. A major shortcoming is the inability to visualize schizont images due to limitations to image in 3D. Here, we were able to overcome such limitation by dissociating differentiated organoids and plating the single cells in a 2D monolayer for 24–48 h prior to infection. Another advantage of HepOrg is the potential to genetically modify the host genome to understand factors involved in parasite development using the recently established protocol of CRISPR/Cas9-based gene knock-in and knock-out[6].

Detailed transcriptomic studies of intracellular *Plasmodium* development in liver cells have been notoriously hard to perform and are restricted to *Plasmodium* species that infect non-human hosts including murine *P. berghei*[34], *P. yoelii*[34,35] and monkey *P. cynomolgi*[36]. These studies used a fluorescently labeled transgenic reporter parasite line to separate infected cells from uninfected cells. Here, we show single-cell RNA-seq data of *Pf*-infected HepOrgs cells obtained on days

4 and 5 p.i. Distinct clusters can be identified that include mature hepatocytes, proliferating hepatocytes and cholangiocyte-like cells in both controls and infected cells. This confirms that a flow sorting-based isolation strategy does not bias selection towards any specific cell type and that the scRNA-seq data is representative of the integral HepOrg cell. The highest number of highly infected cells are found in one particular cluster enriched for mature hepatocytes. This supports the notion that parasite development preferentially takes place in the more differentiated hepatocytes rather than cholangiocyte-like cells[7].

Previously, host transcript changes have been carefully dissected throughout the development of the mouse malaria model, *P. berghei*[34]. In this study, two sequential time points (days 4 and 5 p.i.) were studied with upregulation of genes involved in lipid metabolism and energy generation (glycolysis and gluconeogenesis) comparable with the 12–18 h p.i. time point in the liver-stage development of *P. berghei* (which is completed in 48 h). Another notable change in the liver-stage development of *P. berghei* is the general downregulation of apoptosis machinery and inflammatory response as the parasite develops (12 h p.i. onwards). This was not observed in our dataset but could be due to the non-cancerous nature of HepOrg unlike the hepatoma cell-line Hepa1-6[34].

The finding of hepatic SR-B1 upregulation after Pf infection is novel; previously only associated as a host receptor involved in invasion[16,17] it also has an additional role in the development of rodent malaria models (*P. berghei* and *P. yoelii*) in hepatocyte cell lines. In rodent malaria models, host SR-B1 is the natural determinant or limiting factor for parasite infection. This may be the case for PfNF54 where pre-incubation of HuHeps with BLT1 inhibitors dramatically reduce intracellular parasite numbers, but not for PfNF135 and PfNF175. Single-cell RNA-seq of freshly isolated HuHeps shows that very few cells contain *SCARB1* transcripts (compared to *CD81*, another

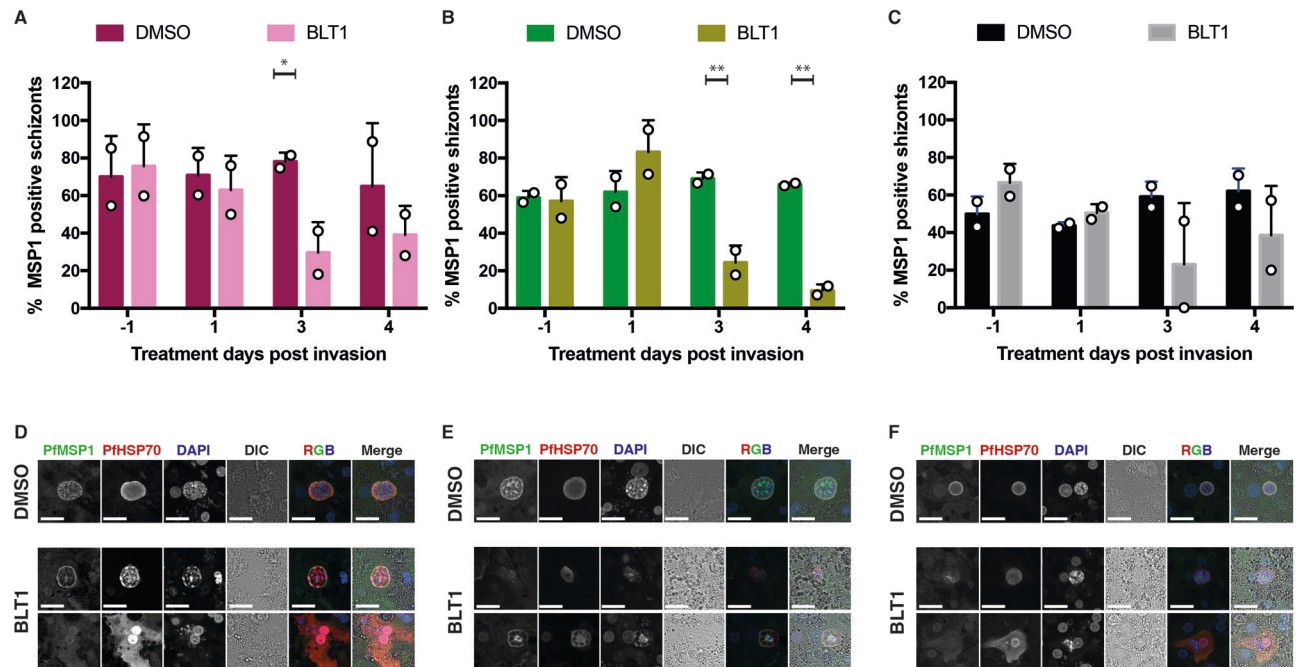

**Fig. 6 | The effect of BLT1 on PfMSP1 development.** The percentage of HuHep Schizonts positive for PfMSP1 expression for NF175 (**A**), NF135 (**B**), and NF54 (**C**). A total of two biological experiments were performed (*n* = 2), each with two technical replicates: at least 100 schizonts were measured from each technical replicate except for NF54 due to low numbers of schizonts surviving. Each bar represents the average and error bars (SD) of two technical replicates (**A**–**C**) from two biological experiments (*n* = 2). Confocal images showing DMSO and BLT1 treated schizonts stained for PfMSP1 and PfHSP70 for NF175 (**D**), NF135 (**E**), and NF54 (**F**) in HuHeps. Scale bar is 25 microns. These images are a representative from two independent experiments. Sidak's multiple comparison test (two sided) is used to compare the DMSO and BLT1 conditions for each parasite strain (**A**–**C**): the *p* values for the significant parameters are annotated (\**p* < 0.05, \*\**p* < 0.01, \*\*\**p* < 0.001). For **A**, *p* value is 0.0488; **B** (left to right), *p* values are 0.0054 and 0.0012.

host invasion receptor) which may explain the relative low infectivity of PfNF54 compared to NF175 and NF135 (unpublished data). While rodent malaria parasite infection may be limited by naturally occurring SR-B1, this is different for *Pf* as there is an upregulation of *SCARB1* transcripts after infection which can be either a host response to infection or active parasite manipulation of host. Unfortunately, we were unable to test the impact of SR-B1 inhibitor on infected HepOrgs due to progressive loss in capacity of PfNF175 to transmit and generate sporozoites[37,38]. Based on our findings, we hypothesize that lipid uptake mediated by SR-B1 is used for the generation of parasite membranes including plasma membranes and/or for the parasitophorous vacuole but also for packaging daughter merozoites: BLT1-treated schizonts are smaller in size and showing atypical MSP-1 staining. The lipids can also be used for host to expand its plasma membrane to accommodate the growing parasites.

In conclusion, human HepOrgs represent a versatile source for in vitro culture that will contribute to better understanding of *Pf*-liver-stage biology. We provide RNA-seq profiles of individual infected cells. A novel role for the hepatic host factor, SR-B1 is identified which reveals heterogeneity among *Pf* strains. The use of human HepOrgs will accelerate clinical development of novel drugs and vaccines. Future studies with other (human) *Plasmodium* species will shed further light on this potential.

## Methods
### Ethics
Human fetal liver samples were obtained at the Leiden University Medical Center (LUMC). The use of samples for purpose of research was approved by the LUMC ethical committee and informed consent was obtained from donors as required.

Primary human liver cells were freshly isolated from remnant surgical material. The samples are anonymized and general approval for use of remnant surgical material was granted in accordance to the Dutch ethical legislation as described in the Medical Research (Human Subjects) Act, and confirmed by the Committee on Research involving Human Subjects, in the region of Arnhem-Nijmegen, the Netherlands.

### Antibodies
**Human hepatocyte organoid culture.** Human hepatocyte organoids (HepOrgs) were established and cultured, essentially as described in Hendriks et al.[6]. Briefly, human hepatocytes were isolated from human fetal tissue by collagenase IV digestion followed by 100 g centrifugation for 5 min. Roughly, 10,000 cells were mixed with Human Hep-Medium (HuHep liver Medium or HLM) and Matrigel[R] (with a ratio of 1:3) and seeded per 24 wells. After solidification, HLM expansion medium (see below) was added and refreshed every 2 days. For the present study, four previously established hepatocyte organoid lines were used (KK2, KK3, KU1, and K1FM).

**HuHep liver medium (HLM) composition.** Human HepOrg liver medium expansion media (HLM-EM) consists of AdDMEM/F12 (Thermo Scientific, with Hepes, GlutaMax and penicillin–streptomycin) plus 15% RSPO1 conditioned medium (Home-made), B27 (without vitamin A), 50 ng/ml EGF(Peprotech), 1.25 mM N-acetyl-L-cysteine (Sigma), 10 nM gastrin (Sigma), 3 mm ChIR99021 (Sigma), 50 ng/ml HGF (Peprotech), 100 ng/ml FGF7 (Peprotech), 100 ng/ml FGF10 (Peprotech), 2 nM A83-01 (Tocris),10 mM Nicotinamide (Sigma), 10 mM Rho Inhibitor Y-27632 (Calbiochem) and 20 ng/ml TGFa.

For fetal HepOrgs, human HepOrg liver medium differentiation media (HLM-DM) was used for organoid differentiation. DM consists of EM plus 10 ng/ml Oncostatin M, 100 nM DAPT and 100 nM Dexamethasone. During culturing, medium was refreshed every 2–3 days. Organoids are usually passaged with the ratio of 1:3–1:4 every 7 days before passage 6, 1:3 every 7–10 days after passage 6. Future distribution of organoids to any third (academic or commercial) party will have to be authorized by the METC UMCU at request of the HUB in

**Table 1 | Reagents needed for immunofluorescence analysis**

| Antibody | Company | Catalog no. | Species | Dilution factor |
|---|---|---|---|---|
| PfHSP70 | StressMarq Biosciences | SPC-186 | Rabbit | 1:75 |
| PfCSP (2A10) | MR4 | 2A10 | Mouse | 1:500 |
| PfEXP2 | European malaria reagent repository | 7.7 | Mouse | 1:1000 |
| PfGAPDH | European malaria reagent repository | 7.2 | Mouse | 1:50000 |
| PfMSP1 | Sanaria and NIH/NIAID | AD233 | Mouse | 1:100 |
| Human Glutamine Synthetase | Abcam | Ab64613 | Mouse | 1:100 |
| Anti-Rabbit Alexa Fluor 594 | Thermofisher Scientific | A11012 | Goat | 1:200 |
| Anti-mouse Alexa Fluor 488 | Thermofisher Scientific | A11029 | Goat | 1:200 |
| DAPI | Thermofisher | D1306 | NA | 1:300 |

order to ensure compliance with the Dutch medical research involving human subjects' act.

**William's B-medium (WLB) composition.** William's B media consists of 1X William's E+ Glutamax (Invitrogen 32551-087) with 100 mM sodium pyruvate (Invitrogen 11360-036), 1% Insulin,transferrin,Selenium (Invitrogen 41400-045), 1% MEM-NEAA (Invitrogen 11140-035), 2% Pen/Strep (Invitrogen 15140-122), 1% Fungizone (GE Healthcare SV30078.01) and 12.8 mM Dexamethosone. Media was filter sterilized and 10% heat inactivated human serum (obtained locally in accordance with ethically approved protocols) was added wherever indicated.

**HLM: WLB composition.** HLM:WLB media consists of equal volumes (1:1) of HLM-DM- and WLB media without serum. Both media should be prepared freshly and mixed.

**HepOrg monolayer plate preparation.** Five organoid lines (KU1, KK2, KK3, K1FM) were maintained in HLM-EM medium. 10–14 days before infection, HepOrgs were transferred to HLM-DM medium. Medium was replenished every 2–3 days with replating in fresh Matrigel around day 7. Falcon 96-well Black/Clear Flat Bottom TC-treated Imaging Microplates (Corning 353219) were coated with collagen coating solution (Sigma 125-50) for 2 days in the incubator. Before cell plating, collagen was neutralized with 150 μl of DMEM. HepOrgs were dissociated into single cells using TrypLE and 50,000 cells/well were plated in the collagen coated plates with 150 μl of HLM. Plated cells were spun down at 3000 rpm for 10 min (acc 9, brake 1) and incubated in the incubator until infection.

**Generation of sporozoites for liver infection.** *Pf* asexual and sexual blood stages were cultures in a semi-automatic system as described in refs. 39–41. *Anopheles stephensi* mosquitoes were reared in the Radboud University Medical Center insectary (Nijmegen, the Netherlands) according to standard operating procedures. Infected mosquitoes were hand-dissected for their salivary glands which were collected in complete William's B medium (William's E medium with Glutamax [Thermo Fisher, 32551-087], supplemented with 1X insulin/transferrin/selenium [Thermo Fisher, 41400-045], 1 mM sodium pyruvate [Thermo Fisher, 11360-070], 1X MEM-NEAA [Thermo Fisher, 11140-035], 2.5 μg/ml Fungizone [Thermo Fisher 15290-018], 200 U/ml penicillin/streptomycin [Thermo Fisher 15140-122] and 1.6 μM dexamethasone [Sigma Aldrich D4902-100MG]) without serum. After homogenization, sporozoites were counted in a Burker-Turk chamber on phase contrast microscope. The sporozoites were supplemented with heat inactivated human sera (HIHS) at 10% of total volume, immediately prior to infection on either HepOrg or HuHeps.

### NF175 infection of HepOrgs
Salivary glands of laboratory-reared infected *A. stephensi* mosquitoes were dissected between days 16 and 21 post-blood meal. For infection

50,000 sporozoites were added to each well suspended in WLB+ 10% Human serum at MOI = 1. Plates were spun at 3000 g for 10 min (acc 9, brake, 1) and incubated at 37 °C for 3 h. Media was replenished after 3 h to remove debris. After initial infection, HepOrg monolayers were maintained in HLM:WLB (1:1) without serum. After infection, media was replenished with HLM: WLB (1:1) every day.

### Primary human hepatocyte infection
Primary human hepatocytes were isolated from patients undergoing elective partial hepatectomy in Yang et al.[7]. Freshly isolated hepatocytes suspended in complete Williams' B media were plated in 96 wells at 62,500 cells per well and kept in a 37 °C (5% CO$_2$) incubator with daily media refreshments. Dissected sporozoites (day 16-21 post blood meal) were added to the HuHeps 48 h after plating at 1:1 ratio in duplicates/triplicates and spun down at 3000 rpm for 10 min on low brakes. Medium is refreshed after 3 h and then on a daily basis. The sporozoite-infected culture was maintained for 5 or 7 days after which the cells were fixed with 4% paraformaldehyde (ThermoFisher Scientific: catalog number 28906) for 10 min. The samples were permeabilized using 1% Triton and stained with the various *P. falciparum* or human antibodies listed above.

For the dose response curve of BLT1 and ITX5061: cryopreserved hepatocytes (two donors: AY40 and AY76) were thawed and infected with either PfNF175 or PFNF135 post-plating. A series of drug concentrations (0, 0.5, 1, 5, 10, 20, 40, 80, 100 μM) were added to infected cultures on day 3 post infection for 48 h (one media refreshment at 24 h). The infected monolayer was fixed on day 7 post infection and processed.

### Viability of uninfected hepatocytes under drug treatment
Cryopreserved hepatocytes are thawed and treated with different concentrations of ITX5061 or BLT1 on either five or six days post plating (i.e. day 3 or day 4 post infection). On day 9 post plating, the viability was determined using the CellTiter-Glo® Luminescent Cell Viability Assay (Promega: #G7570). To test the toxicity of these respective drugs also in HepOrgs (KK2), monolayers were prepared as described above. Three days post plating, HepOrg monolayers were treated with a dose range of ITX5061 or BLT1 for 48 h and on day 7, viability was measured using CellTiter-Glo. Due to variations in parameters such as seeding density or location of the wells in a plate, well-to-well variations can range between 80-120% viability. True toxicity should only be considered for viability less than 80% when compared with control or untreated samples.

**Isolation of blood-stage parasites.** *Pf* asexual stages were cultured in a semi-automatic system as described in Purification of parasitized RBCs was done on a step Percoll gradient consisting of an upper 35% and a lower 65% Percoll layer as shown earlier[42]. After centrifugation, the lower interface contains enriched parasitized RBCs. This interface was collected and diluted in PBS and then sorted using the gates as

used for infected liver cells. As RBCs do not have a nucleus, any granularity in cells (SSC) was considered to be due to infection and the presence of parasite. Method regarding RNA isolation can be found in the method section titled "RNA isolation and qRT-PCR".

**RNA isolation and qRT-PCR.** RNA isolation of organoids, tissues, and primary cells were performed with RNeasy Micro Kit (Qiagen) following the manufacturer's instructions. RNA was reverse transcribed with M-MLV Reverse Transcriptase, RNase H Minus (Promega). qPCR analysis was performed with SYBR Green Mixture (Bio-rad Laboratories) 384 qPCR machine (Bio-rad Laboratories). Primers for qPCR were designed using NCBI Primer-BLAST and listed in Supplemental Table.

**FACS Isolation of infected liver cells.** Organoids were trypsinized into single cells using TryPLE. For monolayers, 96-well cell scrapers (Biotium 22003) was used to detach cells and treated with Accutase (Sigma A6964) for 3 min followed by neutralization with DMEM. Cells were spun at 1200 rpm for 5 min and resuspended in HLM:WLB media. Cell clumps were removed by filtering through a 70-μm filter. Propidium iodide (Thermo Fisher P1304MP) was used for live/dead cell discrimination. Control uninfected cells were used to set FSC and SSC gates. Infected cells were sorted using regular and stringent gates. Stringent gates were set to include cells which had the highest granularity (SSC).

**Single-cell messenger RNA (mRNA) Sequencing.** Single-cell mRNA sequencing (scRNA-seq) was performed using the SORT-seq method[11], which is based on the CEL-Seq2 method[43]. Briefly, single viable cells (i.e. HepOrg-derived cells or RBCs) were directly sorted into wells of a 384-well plate filled with lysis buffer containing well-specific CEL-Seq2 primers. Plates were briefly spun down at $500 \times g$ in a chilled bench-top centrifuge and immediately stored at −80 °C. For library preparation, the stored plates were thawed and heated at 70 °C for 5 min to lyse cells and extract the RNA. The purified RNA was then subjected to first strand and second strand synthesis followed by an overnight in vitro transcription step to generated amplified RNA (aRNA). The aRNA was used as input to generate complementary (cDNA) libraries using Illumina TruSeq primers. Quality control was performed using a Qubit Fluorometer (Thermo Fisher Scientific) and a 2100 Bioanalyzer Instrument (Agilent), as per manufacturer's instructions. All submitted libraries were sequenced on an Illumina NextSeq500 using 75 bp pair-end sequencing with high output (150 million reads per run).

**Bioinformatics analysis.** Reads were mapped to the human GRCh37 genome assembly concatenated with the *P. falciparum* 3D7 strain reference transcriptome (PF3D7), discarding any multi-mapping reads. Read counts were filtered to exclude reads with identical library-, cell- and molecule barcodes and UMI counts were adjusted using Poisson counting statistics[11].

All further bioinformatics analyses were performed in R/Bioconductor environment. scRNA-seq libraries were analyzed using the Seurat v4 package[12]. The initial dataset containing both human and *P. falciparum* 3D7 reads was divided per species and the two resulting datasets were separately analyzed.

For the human dataset, duplicated gene names were made unique using the make.unique function and ERCC92 spike-ins, and total cell transcriptomes with less than 2001 transcript UMIs were removed from the dataset. The remaining transcriptomes were normalized using Seurat's SCTransform function using the vars.to.regress setting to remove confounding sources of variation such as total UMI counts (nCount_RNA), total gene counts (nFeature_RNA) and processed plates. We used Seurat's AddModuleScore function to assess expression of necrosis genes (*FOSB, FOS, JUN, JUNB, ATF3, EGR1, HSPA1A, HSPA1B, HSPB1, IER3, IER2, DUSP1*)[44]. Only cells with a necrosis gene

enrichment score lower than 1 were retained for further analysis. SCTransform with the same settings as before was re-run on the cleaned dataset. Initial cell clustering was generated based on gene expression similarities using Seurat's FindClusters at a resolution = 0.4. The SNN graph was calculated using Seurat's FindNeighbor function and visualized using a script published elsewhere (https://romanhaa. github.io/projects/scrnaseq_workflow/#snn-graph). The cluster tree was generated using Seurat's BuildClusterTree function. Cell cycle analysis was performed using Seurat's CellCycleScoring function. To score gene expression of either hepatocyte marker (*ALB, AFP, RBP4, FABP1, SERPINA1, ASGR2, ASGR1, APOA2, APOC3*) or Cholangiocyte marker (*KRT19, KRT8, KRT18, EPCAM, KRT7*), we used Seurat's AddModuleScore function. Cell cluster marker genes were identified using Seurat's FindAllMarkers function with the following settings: only.pos = TRUE, min.pct = 0.25, thresh.use = 0.25. Expression of lineage marker genes (*ALB, AFP, MKI67, KRT7, EPCAM*) per cell cluster was visualized using Seurat's VlnPlot function set to stack = TRUE. The infection rate of HepOrg cells was defined as follows: none, ≤ 1 *Pf* transcripts; low, 1 <*Pf* transcripts ≤ 50; >50 *Pf* transcripts. Infected HepOrg cells were defined as having more than 1 *Pf* transcript.

For sub-clustering of the dataset from day 5 p.i., the human dataset was split per collection day using Seurat's SplitObject function. After sub-clustering using the same strategy and setting (*e.g.*, resolution = 0.4) as used for the initial clustering, we used the EnrichR package[19] and Seurat's FindMarkers function to, respectively, perform gene-set enrichment analysis (GSEA) and identify differentially expressed (DE) genes (adjusted p value ≤ 0.05) comparing infected cells (*i.e.*, cells with more than 1 total read count assigned to *Plasmodium falciparum* genes) and uninfected cells (*i.e.*, cells with 1 or less total read count assigned to *Plasmodium falciparum* genes). DE genes between infected and uninfected were plotted using Seurat's DoHeatmap function.

For the *P. falciparum* 3D7 dataset, duplicated gene names were made unique using the make.unique function and total cell transcriptomes with less than 50 transcript UMIs were removed from the dataset. Normalization was again performed using Seurat's SCTransform function with the vars.to.regress setting to remove confounding sources of variation such as total UMI counts (nCount_RNA), total gene counts (nFeature_RNA) and processed plates. Clustering was performed as described above using Seurat's FindClusters at a resolution = 0.4 and cluster marker genes were identified using Seurat's FindAllMarkers function with the following settings: only.pos = TRUE, min.pct = 0.25, thresh.use = 0.25. Expression of the TOP 2 genes of each cluster (*i.e.*, the highest expressed genes per cluster) was visualized using Seurat's VlnPlot function set to stack = TRUE. To identify DE genes (adjusted *p* value ≤ 0.05) comparing transcriptomes of the liver stage (*i.e.*, cells collected from infected HepOrgs) and blood stage (*i.e.*, collected infected erythrocytes), we used Seurat's FindMarkers function. DE genes comparing the two stages were plotted using Seurat's DoHeatmap function.

## Microscopy

Leica DMI6000B high content microscope was used for tiling 96 wells for determination of infection rate. For each well, a tile size of 9 × 9 were obtained at 20× objectives. The Zeiss LSM880 with Airyscan at 63× objectives (oil) and 2× zoom or the Leica SP8 SMD at 63× objective (water) were used for obtaining high resolution confocal images.

## Data analysis using FIJI[45]

FIJI version used for the analysis 1.53t

**Infection rate.** See methods section of Yang et al.[7].

**Measurement of schizont size.** Images obtained on the high content microscope were opened in FIJI. Random images were chosen until 100 parasites were measured. Parasites were selected via the region of

interest (ROI) tool using PfHSP70 positivity (red channel) and measured.

**Quantifying intracellular hGS/PfMSP1 by immunofluorescence.** This has been published before[7]. Briefly, for each of the images measured in the previous section, a total background intensity (RawIntDen) in the green channel (as PfMSP-1/hGS is labeled with Alexa-488) was determined using the region of interest (ROI). A background intensity per area (x) was determined (total intensity of whole image divided by area size of the image). A total background intensity for the parasite was determined by using the formula x multiplied by the measured size of the parasite. The measured green intensity within a parasite is determined using the ROI tool to just select the parasite in question. The actual intensity within the parasite is calculated by subtracting the background value from the measured intensity. If the actual intensity is higher than the background value, then the parasite is considered "positive" for hGS/PfMSP1 expression. If lower, than the parasite is "negative". For each condition, more than 50 schizonts were measured except in the cases of BLT1 treatment due to low numbers of surviving schizonts for some Pf lines.

### Reporting summary
Further information on research design is available in the Nature Portfolio Reporting Summary linked to this article.

## Data availability
The sequencing data generated in this study have been deposited in the Gene Expression Omnibus (GEO) under accession number GSE199239. Source data are provided with this paper.

## Code availability
R scripts for the single-cell analysis are available at https://github.com/KaiKretzschmar/HepOrgMalaria.

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

## Acknowledgements

We would like to thank Marga van de Vegte-Bolmer, Rianne Stoter and Wouter Graumans for their excellent work in generating *P. falciparum* parasites and mosquito infections. We would like to thank Jolanda Klaasan, Laura Pelser-Posthumus, Astrid Pouwelsen and Jacqueline Kuhnen for skilfully breeding of mosquitoes and dissection of infected mosquitoes for sporozoites. We would like to thank the Microscopic Imaging Center (MIC) of the Radboud University for access to its facilities. We would like to thank Reinier van den Linden for assistance with flow sorting; Maya Sen, Judith Vivié and Single Cell Discoveries for help with SORT-seq; the Utrecht Sequencing Facility (USEQ) for sequencing and Anko de Graaf and the Hubrecht Imaging Centre (HIC) for help with confocal microscopy. Finally, we would like to thank Judith Bolscher and Marloes de Bruijni (TropIQ Health Sciences) for drug sensitivity data of the Pf strains. A.S.P.Y. is a recipient of a Veni grant from the Dutch Research Council (NWO) talent program (VI.Veni.192.171) and ZonMW Off-Road grant (04510012010050). G.J.v.G. is supported by the European Union's Horizon 2020 research and innovation program under grant agreement No. 733273. This work was supported by an EU/ERC Advanced Grant (to H.C., grant agreement: 67013e) and CRUK grant OPTIMISTICC (to H.C., grant number: C10674/A27140). D.D. was recipient of a VENI grant from the Dutch Research Council (NWO-ALW, 016.Veni.171.015). K.K. was a long-term fellow of the Human Frontier Science Program Organization (HFSPO, LT771/2015), recipient of a VENI grant (NWO-ZonMW, 016.166.140) and is currently funded by the German Cancer Aid (via MSNZ Würzburg/NG3) and an EU/ERC Starting Grant (grant agreement: 101042738). D.H. was supported by a post-doctoral fellowship from the Swedish Society of Medical Research (no. P19-0074) and is supported by a VENI grant from the Dutch Research Council (no. VI.Veni.212.134).

## Author contributions

This study was designed by A.S.P.Y., D.D., K.K., H.C., and R.W.S. The experiments were performed by A.S.P.Y., D.D., K.K., D.H. and Y.v.W. H.H., K.E.B. and S.M.C.d.S.L. assisted with the organoid culture. The data analysis was performed by A.S.P.Y., D.D., K.K., H.C. and D.H. The bioinformatics analysis was performed by K.K. and J.P. J.H.W.d.W. and G.J.v.G. provided the human liver segments and infected mosquitoes respectively. The manuscript was written by A.S.P.Y., D.D., K.K., T.B., H.C., R.W.S. and commented by all co-authors.

## Competing interests

The authors declare no competing interests.
