## [Peer review file · Nature Communications]

REVIEWER COMMENTS

Reviewer #1 (Remarks to the Author):

The manuscript entitled "Development of Plasmodium falciparum liver stages in human fetal hepatocyte organoid cultures" by Yang et al. describes a primary hepatocyte culture method that enables reasonably good infection with *P. falciparum* pre-erythrocytic stages. The authors show that the use of organoid derived fetal hepatocytes can reach up to a 1% infection rate and support later stage liver stage development of *P. falciparum* in vitro. Using this method, the authors transcriptionally profile infected cells versus uninfected cells. They found that *P. falciparum* infectivity is greater in fully differentiated hepatocytes. They also found a previously unrecognized role for SB-R1 in promoting liver stage development.

Major points

The author showed that compared to primary Hu-Hep, cells derived from Hep-Orgs better support late liver stage development. Do Hep-Orgs support the development of merozoites and release of infectious exo-erythrocytic merozoites?

The title implies that hepatocyte organoids are infected. However, the infections are done in 2D culture with cells obtained from organoids. I suggest should be adjusted to clearly reflect this. For example: "Development of Plasmodium falciparum liver stages in human fetal hepatocytes derived from organoids".

The single cell profiling reveals the importance of the host cell metabolism for parasite development. The author found increased expression of SB-R1 and other genes involved in lipid metabolism. The author validated only SB-R1. They should further validate their profiling data by targeting different hits/pathways using chemical inhibition or knockdown. These data will strengthen the biological significance of the manuscript.

To show the role of SB-R1 in the parasite maturation, the authors treat the huHep cultures with the SB-R1 inhibitor BLT-1 at different time of infection. It has been previously shown that SB-R1 is not essential for *P. falciparum* sporozoites entry, that is dependent mostly on CD81. Yang et al., observe reduction in invasion for the NF54 strain but no effect on the other strains NF135 and NF175. The author should comment on the NF54 strain entry blockage observed and how these results are discordant with what has been previously published.

The role of SB-R1 has been studied on the huHep cultures. Why didn't the authors perform the same treatment on their HepOrg cultures where upregulation of SCARB1 has been observed? Furthermore, a better experiment would be knockdown SB-R1 in the HepOrg culture and evaluation of infection.

The methods described in this manuscript could enable the discovery of new therapeutics against *P. falciparum* liver infection. In this regard, a SB-R1 inhibitor called ITX5061 has been tested in clinical trial against HCV infection (Sulkowki et al., 2013, JID; Rowe et al., 2016, Liver Transpl.). The use of the ITX5061 would further validate the role of SB-R1 and provide a translational avenue to this study.

The authors show the infection rates and the mature liver stage schizonts size in four fetal HepOrg lines infected with the NF175 *P. falciparum* strain. In Figure 1A the % of infected cells for the primary human hepatocytes (Huhep) should be included along with the HepOrg lines to maintain consistency throughout the paper.

In supplementary fig S1, the authors show the effect of atovaquone on the different cultures presented in this paper. Concerning the drug treatment, the parasites appeared reduced in size but not in number. Atovaquone seems to be more a cytostatic drug than cytotoxic in this experiment. Did the author after stopping the drug treatment at day 5 follow the culture for longer time periods in order to evaluate the drug efficacy? I suggest also using a different drug such as pyrimethamine.

The authors write they have isolated infected cells by FACS. They should provide images that support this claim. Considering the parasite is not fluorescent how they can be certain that they have sorted infected cells?

Concerning the sc-RNAseq experiment, the author shows higher parasite's reads number in hepatocytes expressing higher level of SCARB1 encoding SR-B1. Did the author evaluate the heterogeneity of the uninfected cells prior to infection? Does the parasite preferentially develop in hepatocytes expressing high SCARB1 or the expression of this gene is induced by the infection?

Minor points

Page 4 line 102 add (Fig. 1B).

Supplementary Fig. S1 panel B, add the legend for HuHep and HepOrg.

Figure S2 Panel A, B and C, the timeline is misleading, add an extra line that shows the time of infection, day 21 corresponds to the *P. falciparum* Day 5 post-infection. Between panel B and C there is Day 21, please remove it and add it to the legend.

A cartoon highlighting the workflow as supplemental image would be beneficial. For example the cartoon could show the steps from the isolation of the fetal hepatocyte, organoid expansion, seeding hepatocyte in 2D and infection.

Reviewer #2 (Remarks to the Author):

This manuscript describes the use of liver organoids (HepOrg) to culture Plasmodium falciparum liver stages. Since there are considerable problems with other methods of culturing liver stages, this does represent a significant and exciting advance in the field. However, the manuscript requires several major revisions and additional experiments.

Major revisions/additional experiments.

1. In some organoid-parasite models, it has been found that the EC50's of inhibitory compounds are shifted higher when organoids are used as the culture system rather than standard 2D culture systems. It would be very informative to see if this is also the case with atovaquone and liver stage Pf. It would also provide more evidence of the utility of this new culture system for drug discovery.

2. It is not clear from the manuscript why, after demonstrating this improved organoid culture system, that the experiments testing effects of BLT1 inhibitor on the parasite were done in primary hepatocytes (HuHeps). It is important to demonstrate the same effects in HepOrgs, especially as this is the point of the paper. Moreover, the line graphs used in Figures 5 and 6 are misleading and exceedingly hard to understand. This data should be shown using straightforward bar graphs, which should also include the standard errors around the mean. From the way the data is presented and the fact that only two biological replicates were performed, it is not clear that these results are statistically significant.

Minor revisions:

Overall, please be sure that all figures are referred to in the text, that there is enough detail in the figure legend that the reader understands how the experiment was done (I often had to plow through the methods to understand some of the figures) and that the figure labels are large enough to be read.

Line 101: This statement seems to refer to Fig 1B (although it is not referenced) . Figure 1B appears to show that schizont size is significantly larger in HepOrgs than HuHeps, but the test in Line 101 says that they are not significantly different. Also in line 102 the test refers to both HepOrg A and B and days 3, 5 and 7, but HepOrg A only goes out to Day 6.

Paragraph starting at line 118: Why were the HepOrgs cultured in 2 D format for the single cell transcriptome analysis? Since it appears that the culture conditions had to be modified to make this work, an explanation of why this approach was taken should be included.

Line 183-184 "day 5 pi onwards" There is no data beyond day 5 in Fig 2 I-J

Lines 186-188. There is no reference describing these parasite strains. Provide either a reference or the data demonstrating the differences in "infectivity and phenotype".

Line 195: "most drastic"? Please provide statistics on differences (see above in major revisions).

Lines 201-211. It is not clear that changes in MSP-1 expression and distribution due to treatment with SRB1 inhibitor translates directly into the conclusion that this is caused by lipid uptake inhibition and inhibition of packaging of daughter merozoites. Were there changes in schizont morphology to support this conclusion? Either soften your conclusions or explain your conclusions more thoroughly.

Lines 274-281: This paragraph is exceedingly confusing. "Rodent model" usually refers to the animal, suggesting in vivo studies. It appears that the authors were using "rodent model" to refer to murine malaria parasites. Please use the correct terms.

Lines 284-285: This statement is missing a reference.

Lines 294-299: Experiments showing reduction of MSP1 do not "directly" show changes in the amount of parasite membrane. Is it not possible that expression of MSP-1 was simply reduced under stress. No experiments were done to show direct effects on parasite membranes. Please soften your conclusions.

Fig 1: Please label proteins as Pf or human consistently throughout the figure legend

Fig 2A: Labels on clusters 2 and 4 are reversed from what they should be.

Figure 2 in general: What were the cut offs for "low" and "high" infection? Were these data collected on day 4 or day 5 post-infection?

Fig 3A: Is this t-SNE map of infected and uninfected cells from a different experiment than that shown in Figure 2A? That pattern looks very different.

Figure 4: It is unclear from the legend what A and B are exactly. Much more detail is needed in the figure legend. Figure 4B is not referred to in the text. There is no description of how single cell transcriptomics were done on blood stage parasites in the methods.

Reviewer #3 (Remarks to the Author):

Comments to authors: The paper by Yang et al describes an effort to utilize their hepatic organoid system to develop an in vitro model, which can be used to analyze the impact of Plasmodium falciparum (Pf) infection on hepatocytes that also can be used for testing new anti-malarial agents. The authors are to be applauded for this effort because of the critical need for new anti-malarial therapies. Since I do not have infectious disease expertise, my comments are restricted to the organoid, scRNA analysis, and drug efficacy aspects of this paper. Also, my comments are intended to encourage the investigators by highlighting those areas where the findings presented in this paper can be strengthened by providing additional data.

Several things should be commented upon by the authors to better clarify the need for and the benefits of using the 3D organoid system. (i) It is noteworthy that the Pf infection rate was between 0.5-1% for all four HepOrg lines, which is significantly lower than that measured in the authors' previous work using primary hepatocytes (3-4%) or in the J7 cell line (3-5%). (ii) The rationale for shifting between using 3D

organoids for cell growth and 2D monolayer cultures in the experiments is not clear. While the cells were obtained from 3D organoid cultures, the experiments were performed using cells grown in 2D cultures (I hope that I am correct in this interpretation. If I am wrong, this could be addressed by adding a diagram to summarize how the experiments were performed to the figures. Is this to enable Pf to have better exposure to the cells, which may be reduced in 3D cultures). In brief, the advantages of using 3D-organoid cultures are not clear?

Their major new finding is that Scavenger Receptor B1 (SRB1) mRNA is upregulated in Pf infected cells in the organoid cultures, which was determined by analysis of scRNA-Seq data. This finding should be strengthened by RT-PCR analysis coupled with antibody staining, which would provide confirmation of the scRNA-Seq findings and new information about whether there is a change in SRB1 protein levels.

The other major finding was that a lipid transport inhibitor (BLT-1) blocked Pf maturation in the hepatic organoid cultures. First, it would be helpful to provide some information about BLT-1 in the text (i.e., described its inhibitory potency, specificity for BLT-1, etc.) to introduce the reader to BLT-1 before presenting the efficacy data. Second, it is imperative to provide information about the BLT-1 concentration ([BLT-1]) used in these studies in the main text and in figures 5-6; and to relate this [BLT-1] to the BLT-1 IC50 for lipid transport. (Within the materials and methods, it was indicated that 20 μ M BLT-1 was used in these studies, while the Nieland et al PNAS paper indicated that 0.01 μ M [BLT-1] inhibited lipid transport in their studies using cultured cells.) Third, as far as I can tell, only a single BLT-1 concentration was tested. The studies should include a dose titration (using 3 different BLT-1 concentrations). Fourth, the effect of serial concentrations of BLT-1 on organoid viability and cellularity (using Calcein AM or equivalent) must be determined to ensure that the effect of this agent on Pf development is not due to organoid toxicity. This toxicity data can then be related to the efficacy data. It is also important to test the effect of several other compounds (other than atovaquone) to provide additional negative controls for the BLT-1 result. At present, only two compounds were tested, and both inhibited Pf growth and/or development. Alternatively, the effect of Pf infection on organoids with a SRB1 KO could be examined.

Minor comments: scRNA-Seq data and other clarifications.

(i) Why is the infection signature in the cholangiocytes similar to that in several hepatocyte clusters? Could the clusters be mislabeled? AFP is not a marker for mature and differentiated hepatocytes (it is a marker for immature cells).

(ii) Clarify how clusters 0, 1, and 3 were distinguished by Seurat. Please also indicate why cluster 2 (cholangiocytes) has high levels of albumin, AFP expression. It is possible that the cholangiocyte and proliferative hepatocyte clusters are mislabeled. Also, why it is the only KI67+ cluster when you indicate that cluster 4 has proliferative hepatocytes?

(iii) It is better to use the Mann Whitney U Test (Wilcoxon Rank Sum Test) for the box/violin plot data because the data does not have a Gaussian (normal) distribution in figs 2C,2I, 2J, 3C, 4D.

(iv) Scale bars should be provided in Figs 1C-D and 6D.

Intentionally signed: Gary Peltz

Reviewer #1 (Remarks to the Author):

The manuscript entitled "Development of *Plasmodium falciparum* liver stages in human fetal hepatocyte organoid cultures" by Yang et al. describes a primary hepatocyte culture method that enables reasonably good infection with *P. falciparum* pre-erythrocytic stages. The authors show that the use of organoid derived fetal hepatocytes can reach up to a 1% infection rate and support later stage liver stage development of *P. falciparum* in vitro. Using this method, the authors transcriptionally profile infected cells versus uninfected cells. They found that *P. falciparum* infectivity is greater in fully differentiated hepatocytes. They also found a previously unrecognized role for SB-R1 in promoting liver stage development.

Major points

The author showed that compared to primary Hu-Hep, cells derived from Hep-Orgs better support late liver stage development. Do Hep-Orgs support the development of merozoites and release of infectious exo-erythrocytic merozoites?

We agree that this would be highly valuable but also be notoriously difficult as shown in previous *in vitro* hepatic studies with *P. falciparum*: individual rings forms have been occasionally seen [1, 2] while multiplying parasites were never recovered. Indeed, we have made several attempts but so far unsuccessful. At this stage, we are therefore unable to conclusively address this point.

The title implies that hepatocyte organoids are infected. However, the infections are done in 2D culture with cells obtained from organoids. I suggest should be adjusted to clearly reflect this. For example: "Development of *Plasmodium falciparum* liver stages in human fetal hepatocytes derived from organoids".

As suggested, we have indeed amended to "Development of *Plasmodium falciparum* liver stages in hepatocytes derived from human fetal liver organoid cultures".

The single cell profiling reveals the importance of the host cell metabolism for parasite development. The author found increased expression of SB-R1 and other genes involved in lipid metabolism. The author validated only SB-R1. They should further validate their profiling data by targeting different hits/pathways using chemical inhibition or knockdown. These data will strengthen the biological significance of the manuscript.

We indeed agree and a logical next step to include other lipid genes of interest. However this will require substantial additional work in a systematic approach and is considered beyond the scope of this paper where just SRB1 is highlighted to show the potential of the organoid model. Validation of individual genes (while indeed worthy) may not be straightforward because: 1) specific inhibitors are often lacking and 2) several candidates are also involved in other important cellular pathways next to lipid metabolism. For example, both ApoB and FASN are also involved in the regulation of cell death pathways (autophagy and apoptosis) known as an important regulator of parasite liver stage development. Therefore, we consider this topic beyond the scope of the current study. We focused on SRB1 as the most upregulated gene with defined inhibitors. As such it showcases the potential translation of HepOrg findings to primary hepatocytes being our major objective.

To show the role of SR-B1 in the parasite maturation, the authors treat the huHep cultures with the SR-B1 inhibitor BLT-1 at different time of infection. It has been previously shown that SR-B1 is not essential for *P. falciparum* sporozoites entry, that is dependent mostly on CD81. Yang et al., observe reduction in invasion for the NF54 strain but no effect on the other strains NF135 and NF175. The author should comment on the NF54 strain entry blockage observed and how these results are discordant with what has been previously published.

The described entry blockage of NF54 is in accordance with published studies by Yalaoui et al ([3] and Rodrigues et al [4]). In these studies, antibodies and inhibitors against SRB1 were added to *in vitro* hepatocytes at 6 to 1 hour prior to NF54 sporozoite invasion resulting in strong reductions of number of intracellular parasites, showing its impact on parasite entry. This contrasts with the results presented by Foquet et al [5] where they showed that the monoclonal antibody (mAB1671) has no impact on Pf liver infection despite having a strong effect in the infection of Hepatitis C virus. This antibody has not been previously (prior to Foquet et al) shown to have an impact on Pf liver infection. Furthermore, it was tested in a humanized liver mice model where bioavailability and degradation of antibodies must be considered. Langlois and colleagues [6] published that SRB1 is not essential for entry in the murine model of malaria parasites, *P. berghei*, which has different developmental kinetics in hepatocytes compared to *P. falciparum*. To conclude, there seems to be no

consensus as to whether SRB1 is essential for PfNF54 invasion and our data shown here is in accordance with some previously published data.

The role of SR-B1 has been studied on the huHep cultures. Why didn't the authors perform the same treatment on their HepOrg cultures where upregulation of SCARB1 has been observed? Furthermore, a better experiment would be knockdown SR-B1 in the HepOrg culture and evaluation of infection.

The previous SR-B1 studies in *Plasmodium falciparum* infected HuHep cultures [4] aimed to address parasite entry (i.e. 1 hour prior to the addition of parasite). Here, we show the profound impact on parasite development which has not been shown previously.

The possible importance of SRB1 was discovered in our HepOrg scRNAseq profile. Rather than using the HepOrgs for phenotypic studies as suggested by the reviewer, we decided to directly use the more biological relevant primary human hepatocytes to make a convincing case.

We did, however, made a serious effort to accommodate the reviewer's comment. Unfortunately, after long periods of intense efforts we had to conclude that NF175 cultures had lost the capacity to induce infected mosquitoes and production of sporozoites. Consequently, we are unfortunately unable to conduct the experiment to generate the requested data. Irrespectively, we hope that the SRB1 example as a novel finding will sufficiently illustrate the value and the potential of the HepOrgs model.

As for knockdown approaches, we see technical limitations for our specific study purposes. Although an attractive option, the use of siRNA has so far only been tested in liver stages at 48 hours prior to parasite infection [4]. Possible effects of siRNA interference do require sufficient time for cells to recover from that intervention (transfection/electroporation siRNA) [7]. However, given the timeframe of our study interest of SR-B1 (96 hours post-infection), siRNA will have to be applied almost concurrently with the sporozoites, which will trigger major cellular insults affecting hepatocyte health and ability of the parasites to properly develop into schizonts. Given these considerations, we have decided not to make use of this approach.

The methods described in this manuscript could enable the discovery of new therapeutics against *P. falciparum* liver infection. In this regard, a SB-R1 inhibitor called ITX5061 has been tested in clinical trial against HCV infection (Sulkowki et al., 2013, JID; Rowe et al., 2016, Liver Transpl.). The use of the ITX5061 would further validate the role of SB-R1 and provide a translational avenue to this study.

We thank the reviewer for this valuable suggestion. First, we tested a dose response curve on ITX5061 in two donors of cryopreserved primary human hepatocytes (AY40 and AY76), showing that, ITX5061 does not affect parasite development when applied on day 3 or 4 post infection (for a 48-hour period) in contrast to the well-accepted SR-B1 inhibitor BLT1. Next we conducted a more extensive dose response curve and show that ITX5061 is active at concentrations above 40 μ M. The difference in potency between ITX5061 and BLT1 could be the dual function of ITX5061 which is both an antagonist for SRB1 and p38 MAP kinase. The latter may possible also impact on parasite development but is unknown. The data have been incorporated in the manuscript Figure S7 and the data referred to on line 185 to 189.

The authors show the infection rates and the mature liver stage schizonts size in four fetal HepOrg lines infected with the NF175 *P. falciparum* strain. In Figure 1A the % of infected cells for the primary human hepatocytes (HuHep) should be included along with the HepOrg lines to maintain consistency throughout the paper.

We have adopted this point and added the sentence (line 99) " This is lower than the published infection of PfNF175 in freshly isolated primary human hepatocytes[8]." However, unpublished data shows that PfNF175 in commercially cryopreserved primary human hepatocytes to be 0.5-2% (which is within the range of HepOrg).

In supplementary fig S1, the authors show the effect of atovaquone on the different cultures presented in this paper. Concerning the drug treatment, the parasites appeared reduced in size but not in number. Atovaquone seems to be more a cytostatic drug than cytotoxic in this experiment. Did the author after stopping the drug treatment at day 5 follow the culture for longer time periods in order to evaluate the drug efficacy? I suggest also using a different drug such as pyrimethamine.

Atovaquone activity seems indeed to be primarily cytostatic in infected primary human hepatocytes showing sharp decreases in parasite size rather than in parasite numbers. In contrast, this is clearly not the case for infected HepOrgs where we show both a significant drop in numbers down to almost zero and a decrease in

schizont size (Figure S1). The infected HepOrg cultures were not followed beyond day 5 post-infection to the point that disappearance of parasites may also be explained by natural rupture of mature schizonts.

Indeed pyrimethamine might be an interesting alternative drug to test with known liver stage activity. We have added extra panels to Figure S1 and added accompany text in the main manuscript (lines 113-114). However, when tested in primary hepatocytes NF175 is not sensitive to pyrimethamine as illustrated in the figure below (Panel C - atovaquone and D – pyrimethamine). Panel E shows dose response in infected HepOrgs with both drugs corroborating findings in primary hepatocytes.

The authors write they have isolated infected cells by FACS. They should provide images that support this claim. Considering the parasite is not fluorescent how they can be certain that they have sorted infected cells? This is a valid point and it was indeed a challenge to find a way to isolate infected cells without labelled parasites. Our working hypothesis was based on discriminatory differences in granularity between infected cells containing daughter parasites (side scatter or SSC) and uninfected cells. Cells were sorted primarily based on high cut off for granularity (stringent versus regular) (see Figure S2F). Figure S3G supports our assumption; cells sorted with stringent gates were found have a higher infection status (with the readout being parasite transcripts) as compared to regular gates. Since cells were directly sorted into lysis buffer in the scRNA-seq experiments, imaging was not possible.

Concerning the sc-RNAseq experiment, the author shows higher parasite's reads number in hepatocytes expressing higher level of SCARB1 encoding SR-B1. Did the author evaluate the heterogeneity of the uninfected cells prior to infection? Does the parasite preferentially develop in hepatocytes expressing high SCARB1 or the expression of this gene is induced by the infection?

We thank the reviewer for this valid comment. We have now evaluated the SCARB1 expression in control cells from HepOrgs and compared these with SCARB1 expression in Pf-exposed (sample) organoids (Supplementary Figure S6D). SCARB1 expression is robustly lower in control cells compared to Pf-exposed organoids. In addition, the SCARB1 was more heterogeneously expressed in the Pf-exposed HepOrgs than in the control condition. From our subsequent inhibitor experiments, we further conclude that SCARB1/SR-B1 affects parasite growth. Whether Pf preferentially infects SCARB1-high cells or induces SCARB1 expression will need further testing.

Minor points

Page 4 line 102 add (Fig. 1B).

This has been added.

Supplementary Fig. S1 panel B, add the legend for HuHep and HepOrg.

This has been added.

Figure S2 Panel A, B and C, the timeline is misleading, add an extra line that shows the time of infection, day 21 corresponds to the P. falciparum Day 5 post-infection. Between panel B and C there is Day 21, please remove it and add it to the legend.

This has been amended accordingly.

A cartoon highlighting the workflow as supplemental image would be beneficial. For example the cartoon could show the steps from the isolation of the fetal hepatocyte, organoid expansion, seeding hepatocyte in 2D and infection.

This has been addressed in Panel A of Figure S2.

Reviewer #2 (Remarks to the Author):

This manuscript describes the use of liver organoids (HepOrg) to culture Plasmodium falciparum liver stages. Since there are considerable problems with other methods of culturing liver stages, this does represent a significant and exciting advance in the field. However, the manuscript requires several major revisions and additional experiments.

Major revisions/additional experiments.

1. In some organoid-parasite models, it has been found that the EC50's of inhibitory compounds are shifted higher when organoids are used as the culture system rather than standard 2D culture systems. It would be very informative to see if this is also the case with atovaquone and liver stage Pf. It would also provide more evidence of the utility of this new culture system for drug discovery.

We have been unable to trace such published drug findings in “organoid-parasite models” as referred to by the reviewer. Irrespectively we do confirm such a shift as presented in the supplementary figure 1: the EC50 of atovaquone in HepOrgs with a significant decrease in infection rate at 10nM is absent in the primary human hepatocytes.

2. It is not clear from the manuscript why, after demonstrating this improved organoid culture system, that the experiments testing effects of BLT1 inhibitor on the parasite were done in primary hepatocytes (HuHeps). It is important to demonstrate the same effects in HepOrgs, especially as this is the point of the paper. Moreover, the line graphs used in Figures 5 and 6 are misleading and exceedingly hard to understand.

This data should be shown using straightforward bar graphs, which should also include the standard errors around the mean. From the way the data is presented and the fact that only two biological replicates were performed, it is not clear that these results are statistically significant.

The possible importance of SRB1 was discovered in our HepOrg scRNAseq profile. Rather than using the HepOrgs for phenotypic studies, we decided to directly use the more biological relevant primary human hepatocytes to make a convincing case.

We did, however, adopt the reviewer’s comment. Unfortunately, after long periods of intense efforts we had to conclude that NF175 cultures had lost the capacity to induce infected mosquitoes and production of sporozoites. Consequently, we are unfortunately unable to conduct the experiment to generate the requested data. Irrespectively, we hope that the SRB1 example will sufficiently illustrate the value and potential HepOrgs as a novel model for the gold standard primary human hepatocytes.

We have amended the graphs in question and attached performed Sidak’s multiple comparison test. Statistically significant parameters are annotated.

Minor revisions:

Overall, please be sure that all figures are referred to in the text, that there is enough detail in the figure legend that the reader understands how the experiment was done (I often had to plow through the methods to understand some of the figures) and that the figure labels are large enough to be read.

Line 101: This statement seems to refer to Fig 1B (although it is not referenced) . Figure 1B appears to show that schizont size is significantly larger in HepOrgs than HuHeps, but the test in Line 101 says that they are not significantly different. Also in line 102 the test refers to both HepOrg A and B and days 3, 5 and 7, but HepOrg A only goes out to Day 6.

We do apologize for the confusion. We have added the reference to Fig 1B on line 103. HepOrg A and B (as now stated in the figure legends) represent two different experiments each with four different organoid lines. HepOrg A was followed from day 3-6 post infection while HepOrg B from day 5-8 post infection. The combined data suggest that size is not clearly different between HuHep and HepOrgs.

Paragraph starting at line 118: Why were the HepOrgs cultured in 2 D format for the single cell transcriptome analysis? Since it appears that the culture conditions had to be modified to make this work, an explanation of why this approach was taken should be included.

We do again apologize for the confusion: we have provided/amended a schematic representation of the infection protocol in Supplementary Fig 2A. We had initially pursued microinjection of sporozoites to HepOrgs in a 3D format but were unsuccessful in obtaining an infection, however 2D format did result in an infection. This could be due to the compact nature of HepOrgs which differ from those generated from cholangiocytes[9] with big lumen.

Line 183-184 “day 5 pi onwards” There is no data beyond day 5 in Fig 2 I-J

We have removed the word “onwards”.

Lines 186-188. There is no reference describing these parasite strains. Provide either a reference or the data demonstrating the differences in “infectivity and phenotype”.

We have provided reference 7.

Line 195: “most drastic”? Please provide statistics on differences (see above in major revisions).

We have amended (see point under major revisions) and have removed the “most drastic on when infected HuHeps were treated day 3 and 4 p.i.”.

Lines 201-211. It is not clear that changes in MSP-1 expression and distribution due to treatment with SRB1 inhibitor translates directly into the conclusion that this is caused by lipid uptake inhibition and inhibition of packaging of daughter merozoites. Were there changes in schizont morphology to support this conclusion? Either soften your conclusions or explain your conclusions more thoroughly.

We have softened that conclusion which now reads “The combined data of reduced schizont size and lack of or atypical MSP-1 staining pattern suggest that SR-B1 may be involved in the maturation of liver schizonts.”

Lines 274-281: This paragraph is exceedingly confusing. “Rodent model” usually refers to the animal, suggesting in vivo studies. It appears that the authors were using “rodent model” to refer to murine malaria parasites. Please use the correct terms.

We have amended the sentence to read “Detailed transcriptomic studies of intracellular Plasmodium development in liver cells have been notoriously hard to perform and are restricted to Plasmodium species that infect non-human hosts...” (line 263 to 265).

Lines 284-285: This statement is missing a reference.

We have added the reference and changed the sentence for clarity.

Lines 294-299: Experiments showing reduction of MSP1 do not “directly” show changes in the amount of parasite membrane. Is it not possible that expression of MSP-1 was simply reduced under stress. No experiments were done to show direct effects on parasite membranes. Please soften your conclusions.

We agree that a reduction of MSP1 does not directly show changes in the amount of parasite membranes. In the manuscript, we measured the size of the schizonts (along with MSP1 positivity) and that is an indirect measure of the amount of parasite membrane/parasitophorous vacuole. We have softened our text to “Based on our findings, we hypothesize that lipid uptake mediated by SR-B1 is used for the generation of parasite membranes including plasma membranes and/or for the parasitophorous vacuole but also for packaging daughter merozoites: BLT-1-treated schizonts are smaller in size and showing atypical MSP1 staining.”

Fig 1: Please label proteins as Pf or human consistently throughout the figure legend

We have amended accordingly.

Fig 2A: Labels on clusters 2 and 4 are reversed from what they should be.

Response: Indeed, cluster 4 and 2 have been mis-labelled in the submitted manuscript. We do apologize for this error that now has been corrected in the revised manuscript.

Figure 2 in general: What were the cut offs for “low” and “high” infection? Were these data collected on day 4 or day 5 post-infection?

The infection rates were defined based on the number of Pf transcripts: none, less than or equal to 1 Pf transcripts; low, between 1 and less than or equal to 50 Pf transcripts; and high, more than 50 transcripts. This information is now added directly into Figure 2D for clarity. Furthermore, this explanation is provided in the bioinformatics analysis section of the methods. All data presented in Fig. 2 and associated figures were collected on day 4 and 5 post-infection.

Fig 3A: Is this t-SNE map of infected and uninfected cells from a different experiment than that shown in Figure 2A? That pattern looks very different.

The t-SNE map shown in Fig. 2A represents the data collected on day 4 and 5 post-infection, while the t-SNE map in Fig. 3A was generated from restricting the analysis to the data collected on day 5 post-infection (via dataset sub-clustering, see Methods, lines 506-514). Since organoid-derived cells collected on day 5 post-infection showed higher infection rates, we were thereby able to get a better picture on the differences between highly infected and non-infected cells.

Figure 4: It is unclear from the legend what A and B are exactly. Much more detail is needed in the figure legend.

Figure 4B is not referred to in the text. There is no description of how single cell transcriptomics were done on blood stage parasites in the methods.

This has been amended and details on the experimental description have been added in the methods.

Reviewer #3 (Remarks to the Author):

Comments to authors: The paper by Yang et al describes an effort to utilize their hepatic organoid system to develop an in vitro model, which can be used to analyze the impact of Plasmodium falciparum (Pf) infection on hepatocytes that also can be used for testing new anti-malarial agents. The authors are to be applauded for this effort because of the critical need for new anti-malarial therapies. Since I do not have infectious disease expertise, my comments are restricted to the organoid, scRNA analysis, and drug efficacy aspects of this paper. Also, my comments are intended to encourage the investigators by highlighting those areas where the findings presented in this paper can be strengthened by providing additional data.

Several things should be commented upon by the authors to better clarify the need for and the benefits of using the 3D organoid system. (i) It is noteworthy that the Pf infection rate was between 0.5-1% for all four HepOrg lines, which is significantly lower than that measured in the authors' previous work using primary hepatocytes (3-4%) or in the J7 cell line (3-5%).

The fetal hepatocyte organoid system has several advantages over conventional primary human hepatocyte cultures and represents a complementary, new Pf model system. Although infection rates may be generally lower, the organoids can be expanded and maintained for longer periods of time. In addition, future studies could integrate CRISPR genome editing with the organoids, to establish knock-out lines or fluorescent reporter knock-in lines that could further elucidate Pf biology at precision gene level. (Lines 256-259)

With regard to the notion that experiments were performed in 2D cultures, this is discussed in lines 256-259. Furthermore, we have attempted to infect 3D HepOrgs through microinjection but was unsuccessful.

We apologize for the confusion but all HepOrg infections were performed on differentiated hepatocytes plated out in a 2D format (rather than a 3D system). It is indeed noteworthy that the Pf infection rate of NF175 is lower in all four HepOrg lines compared to primary hepatocytes. A plausible explanation may be that the final differentiation stages of the HepOrg cells may deviate from the gold-standard primary hepatocytes.

With regards to the J7 cell lines [10], we have some reservations with regard to the potential suitability of these cells : 1) there is no representative image of day five post infection Pf schizont in the paper, and 2) its citations have only been used in reviews [11-15] without any published follow-up studies. This is highly remarkable for a much-needed hepatocyte cell-line with characteristics that can seemingly outperform even the gold-standard primary hepatocytes.

(ii) The rationale for shifting between using 3D organoids for cell growth and 2D monolayer cultures in the experiments is not clear. While the cells were obtained from 3D organoid cultures, the experiments were performed using cells grown in 2D cultures (I hope that I am correct in this interpretation. If I am wrong, this could be addressed by adding a diagram to summarize how the experiments were performed to the figures. Is this to enable Pf to have better exposure to the cells, which may be reduced in 3D cultures). In brief, the advantages of using 3D-organoid cultures are not clear?

A diagram is provided as to the scheme of infecting HepOrg (see Supplementary Figure 2A).

Their major new finding is that Scavenger Receptor B1 (SRB1) mRNA is upregulated in Pf infected cells in the organoid cultures, which was determined by analysis of scRNA-Seq data. This finding should be strengthened by RT-PCR analysis coupled with antibody staining, which would provide confirmation of the scRNA-Seq findings and new information about whether there is a change in SRB1 protein levels.

We have previously tested a published SRB1 antibody [16] in primary human hepatocytes. However, we found that the signal was too variable for us to make a conclusive statement of the number of SRB1 positive cells nor the amount of SRB1 signals present within the host cell. Due to the low yield of infected cells, standard RT-PCR or bulk RNA-sequencing protocols are not feasible that is why we decided to use single-cell transcriptomics. Furthermore, several studies have shown very high correlation between qPCR and RNA-seq data[17, 18]. RT-PCR analysis therefore will add little extra utility to our data. However, to strengthen our SRB1 data, we have now also confirmed (in addition to BLT1) that another antagonist of SRB1 (ITX5061) also impacts on parasite numbers when treated during infection (Figure S7).

The other major finding was that a lipid transport inhibitor (BLT-1) blocked Pf maturation in the hepatic

organoid cultures. First, it would be helpful to provide some information about BLT-1 in the text (i.e., described its inhibitory potency, specificity for BLT-1, etc.) to introduce the reader to BLT-1 before presenting the efficacy data.

We have amended the sentence (lines 186-187) and have provided two references where BLT1 has been previously used before in the context of malaria.

Second, it is imperative to provide information about the BLT-1 concentration ([BLT-1]) used in these studies in the main text and in figures 5-6; and to relate this [BLT-1] to the BLT-1 IC50 for lipid transport. (Within the materials and methods, it was indicated that 20 uM BLT-1 was used in these studies, while the Nieland et al PNAS paper indicated that 0.01 uM [BLT-1] inhibited lipid transport in their studies using cultured cells.)

Rodrigues and colleagues (Figure 4B; [4]) have performed dose-response curves with mouse malaria parasites and based our experiments on these findings. This paper is now cited in the methods section (line 412). We did perform a dose response curve with BLT-1 in Pf-infected hepatocytes (see reviewer 1, point 6).

Third, as far as I can tell, only a single BLT-1 concentration was tested. The studies should include a dose titration (using 3 different BLT-1 concentrations). Fourth, the effect of serial concentrations of BLT-1 on organoid viability and cellularity (using Calcein AM or equivalent) must be determined to ensure that the effect of this agent on Pf development is not due to organoid toxicity. This toxicity data can then be related to the efficacy data. It is also important to test the effect of several other compounds (other than atovaquone) to provide additional negative controls for the BLT-1 result.

We have addressed the reviewers comment by providing a dose response curve of BLT1 in primary hepatocytes see above. Furthermore, we also stained the treated primary culture for cleaved Caspase 3 (Cell Signaling Technology; #9661) and did not find any positive cells for cleaved caspase 3 (indication for host cell death) even at the highest concentration of BLT1 (data not shown). Additionally, we have also performed CellTiter-Glo (Promega) on treated uninfected hepatocytes for both BLT1 and ITX5061 and show that there is no significant loss of cell health compared to the DMSO control (Figure S7A).

At present, only two compounds were tested, and both inhibited Pf growth and/or development. Alternatively, the effect of Pf infection on organoids with a SRB1 KO could be examined.

The novelty of our SR-B1 findings is its involvement in parasite development rather than cell entry. Creating an SR-B1 knockout in HepOrg would prohibit studies on this novel role as the parasite will be unable to enter the host cell.

Minor comments: scRNA-Seq data and other clarifications.

(i) Why is the infection signature in the cholangiocytes similar to that in several hepatocyte clusters? Could the clusters be mislabeled? AFP is not a marker for mature and differentiated hepatocytes (it is a marker for immature cells).

Indeed, cluster 4 and 2 were mis-labelled (swapped) in the submitted manuscript. We do apologize for this error that now has been corrected in the revised manuscript. We included AFP as a marker for hepatocytes, since these hepatocyte organoids are of fetal origin. Thus, in this case AFP expression does not mean dedifferentiation, but actually marks proper hepatocytes. These cells also always express ALB and other mature markers. However, we do agree that marking mature hepatocytes as AFP+ is confusing to the field, and therefore now removed AFP to identify the mature cluster.

(ii) Clarify how clusters 0, 1, and 3 were distinguished by Seurat. Please also indicate why cluster 2 (cholangiocytes) has high levels of albumin, AFP expression. It is possible that the cholangiocyte and proliferative hepatocyte clusters are mislabeled. Also, why it is the only KI67+ cluster when you indicate that cluster 4 has proliferative hepatocytes?

In supplementary table 1, we provide a list of differentially expressed genes specific for each cell cluster. The TOP 5 genes for cluster 0 are, for example, VIM, FILIP1L, RRAS2, TES and MYH9. While cluster 1 is enriched in transcripts of FGG, STOM, MBNL3, GC and SRPINF1. The TOP 5 genes for cluster 3 are SLC22A7, HP, CYP8B1, AGXT2 and UROC1. Otherwise, indeed, as mentioned, the labels for cluster 2 and 4 were accidentally swapped. This error has now been corrected.

iii) It is better to use the Mann Whitney U Test (Wilcoxon Rank Sum Test) for the box/violin plot data because

the data does not have a Gaussian (normal) distribution in figs 2C,2I, 2J, 3C, 4D.

This has now been done in both main and supplementary figures, where applicable.

(iv) Scale bars should be provided in Figs 1C-D and 6D.

Visible scale bars have now been provided.

Intentionally signed: Gary Peltz

1. Mazier, D., et al., *Complete development of hepatic stages of Plasmodium falciparum in vitro*. Science, 1985. **227**(4685): p. 440-2.
2. Roth, A., et al., *A comprehensive model for assessment of liver stage therapies targeting Plasmodium vivax and Plasmodium falciparum*. Nat Commun, 2018. **9**(1): p. 1837.
3. Yalaoui, S., et al., *Scavenger receptor BI boosts hepatocyte permissiveness to Plasmodium infection*. Cell Host Microbe, 2008. **4**(3): p. 283-92.
4. Rodrigues, C.D., et al., *Host scavenger receptor SR-BI plays a dual role in the establishment of malaria parasite liver infection*. Cell Host Microbe, 2008. **4**(3): p. 271-82.
5. Foquet, L., et al., *Anti-CD81 but not anti-SR-BI blocks Plasmodium falciparum liver infection in a humanized mouse model*. J Antimicrob Chemother, 2015. **70**(6): p. 1784-7.
6. Langlois, A.C., et al., *Plasmodium sporozoites can invade hepatocytic cells independently of the Ephrin receptor A2*. PLoS One, 2018. **13**(7): p. e0200032.
7. Han, H., *RNA Interference to Knock Down Gene Expression*. Methods Mol Biol, 2018. **1706**: p. 293-302.
8. Yang, A.S.P., et al., *Zonal human hepatocytes are differentially permissive to Plasmodium falciparum malaria parasites*. EMBO J, 2021: p. e106583.
9. Huch, M., et al., *Long-term culture of genome-stable bipotent stem cells from adult human liver*. Cell, 2015. **160**(1-2): p. 299-312.
10. Tweedell, R.E., et al., *The Selection of a Hepatocyte Cell Line Susceptible to Plasmodium falciparum Sporozoite Invasion That Is Associated With Expression of Glypican-3*. Front Microbiol, 2019. **10**: p. 127.
11. Ramirez-Flores, C.J., et al., *Transcending Dimensions in Apicomplexan Research: from Two-Dimensional to Three-Dimensional In Vitro Cultures*. Microbiol Mol Biol Rev, 2022. **86**(2): p. e0002522.
12. Valenciano, A.L., et al., *In vitro models for human malaria: targeting the liver stage*. Trends Parasitol, 2022.
13. Arez, F., et al., *Bioengineered Liver Cell Models of Hepatotropic Infections*. Viruses, 2021. **13**(5).
14. Loubens, M., et al., *Plasmodium sporozoites on the move: Switching from cell traversal to productive invasion of hepatocytes*. Mol Microbiol, 2021. **115**(5): p. 870-881.
15. Mukherjee, P., G. Burgio, and E. Heitlinger, *Dual RNA Sequencing Meta-analysis in Plasmodium Infection Identifies Host-Parasite Interactions*. mSystems, 2021. **6**(2).
16. Pewkliang, Y., et al., *A novel immortalized hepatocyte-like cell line (imHC) supports in vitro liver stage development of the human malarial parasite Plasmodium vivax*. Malar J, 2018. **17**(1): p. 50.

17. Griffith, M., et al., *Alternative expression analysis by RNA sequencing*. Nat Methods, 2010. **7**(10): p. 843-7.
18. Wu, A.R., et al., *Quantitative assessment of single-cell RNA-sequencing methods*. Nat Methods, 2014. **11**(1): p. 41-6.

REVIEWER COMMENTS

Reviewer #1 (Remarks to the Author):

thank you for thoughtfully addressing all of my comments.

Reviewer #3 (Remarks to the Author):

The clarity of the paper has been markedly improved by changing the title and by the addition of Fig. S2A. It is now clear that the cultures analyzed were 2-D hepatocyte cultures that were derived from organoid cultures. Also, the conclusions are strengthened by addition of the BLT1 dose-response and toxicity data; by showing the effect of a 2nd SRB1 inhibitor (ITX5061); and by correcting the cluster mislabeling. However, there are 2 remaining areas that could improve the paper:

1. Demonstrating that a 2nd SRB1 antagonist is active strengthens the finding that SRB1 has a role in Pf development. However, since up regulation of SRB1 expression after infection is a major finding presented in this paper, it still requires confirmation by a method other than scRNA-Seq. I am not sure why cell sorting cannot be done with analysis of infected cells; but could in situ hybridization or some other technique be used for analyzing for Pf and SRB1 co-expression. If this is too difficult, I understand.

2. Since the analyses use 2D-organ derived hepatocyte, why are primary hepatocytes used for toxicity testing instead of the organ derived hepatocytes? To be useful for drug screening, efficacy and toxicity should be tested in the same system with the same cells.

Intentionally Signed by Gary Peltz

Reviewer #4 (Remarks to the Author):

Reviewer #2

Overall, the authors addressed most minor concerns and some major concerns of this reviewer.

Have authors addressed major concerns of this reviewer?

1. This reviewer was concerned that the organoid model would yield significantly different inhibitory concentrations of atovaquone between the HuHep and HepOrg models based on previously published organoid-parasite systems (See Funkhouser-Jones et al 2020). The reviewer suggested that the authors include dose-response curves and EC50 of atovaquone in infected HuHeps and HepOrgs. The author's included a dose-response curve and EC50 of atovaquone only in HuHeps and not for HepOrgs (Fig.S1C). The authors did express that they observed a "significant decrease" of atovaquone at 10 nM in the HepOrgs that was not observed in the HuHeps (Fig.S1A). However, since it is unclear that whether this experiment was performed with three biological replicates and there is no statement regarding statistical analysis for these data, the claim of significant differences in the effect of atovaquone on parasite inhibition in both cell culture models remains unclear. Therefore, this concern was not addressed by the authors and requires further investigation.

2. This reviewer was concerned as to why the authors did not use the HepOrg model to explore the effects of the SR-B1 inhibitor: BLT1, and the role of SR-B1 in parasite development. The authors rebuttal expressed that they used HuHep because they are a "more biological relevant" cell line. Simply, the authors performed scRNAseq on HepOrgs, found differential expression of lipid metabolism related genes (including SR-B1) between infected and uninfected HepOrgs, and used HuHep for

phenotypic studies to support their hypothesis of SR-B1's involvement for parasite development in hepatocytes. Based on this, it appears that while the authors developed a successful organoid model for parasite infection and transcriptomic analyses, the authors encountered technical limitations whereby phenotypic studies were unable to be conducted in the HepOrg system. There is no sentence in the manuscript that explicitly justifies or explains the use of HuHeps for the phenotypic studies instead of the HepOrgs. Therefore, this concern was not addressed by the authors. However, the use of HuHeps to complement the scRNAseq findings provide useful data that sustains the authors' hypothesis.

3. This reviewer was concerned with the data visualization adopted in Figures 5 and 6. The authors chose to visualize the data using dots connected by lines, and this reviewer thought that it was a misrepresentation of the data. This reviewer suggested that the data should be represented in bar graphs depicting the means and their error bars. The authors followed this reviewer's suggestion and implemented bar graphs in these figures and included statistical analyses. However, the figure legends were not edited to reflect the changes and still give the impression that there are dots in the graph (i.e. Legend of Figure 5 says: "Each dot represents a biological experiment", while there are no dots in the graph.)

Have authors addressed minor concerns of this reviewer?

1. The overall concern was that all figures were referred to in the text, that there was enough detail in the figure legend to understand the experiment and that the figure labels are large enough to be read. Each figure is now referred to in the text, the figure legends adequately explain the figure (although more detail would be useful), and the figure labels are an appropriate size. This concern was addressed.

2. There were concerns in lines 101 and 102. In line 101 the statement is that the size of schizonts is not significantly different between HuHeps and HepOrgs according to Fig1B. In the figure, there is an apparent difference in schizont size, and, although the authors state that there is no significant difference, no statistical test is mentioned to support this conclusion. In line 102, it was not clear that HepOrg A and HepOrg B were to different experiments. The authors corrected this and is now explicitly stated. This concern was partially addressed.

3. Paragraph starting in line 118 lacked an explanation as to why the authors decided to use the HepOrgs in 2D format to conduct the transcriptome analysis. The authors now have an explanation for this approach. This concern was addressed.

4. Line 183-184 had a misleading word. The authors removed it. This concern was addressed.

5. Line 186-188 was missing a reference. The authors included it. This concern was addressed.

6. Line 195 had a misleading phrase and lacked statistical test to support the differences observed with BLT-1. The authors rephrased the line and included statistical tests. This concern was addressed.

7. Lines 201-211 stated conclusions that did not directly translate from the experiments on MSP1 expression as a result of SRB1 inhibition. The authors softened the conclusions based on the reviewer's comments. This concern was addressed.

8. In lines 274-281 the authors referred to a "rodent model" to illustrate findings from past studies with Plasmodium species that infect murine organisms. The use of "rodent model" is confusing because it suggests the use of in vivo studies. This reviewer suggested to use the correct terms. While the authors address the concern of the misleading phrase in these lines, the use of "rodent model" can still be read throughout the manuscript (e.g. Lines 204 and 282). Thus, the authors partially addressed this concern.

9. The statement on lines 284-285 was missing a reference. The authors addressed this concern and included a reference for the statement.

10. Lines 294-299 stated conclusions that did not directly translate from the experiments on MSP1 expression as a result of SRB1 inhibition. The authors softened the conclusions based on the reviewers' comments. This concern was addressed.

11. Figure 1 labels were not consistent throughout the legend. The authors addressed this concern and the labels are consistent.

12. Figure 2A was mislabeled. The authors addressed this concern and revised the labeling of the figure.

13. In figure 2, it was unclear 1) what were the cutoffs of infection for the categorization of "low" and "high" parasite transcripts and 2) at what day post infection the data was collected. The authors addressed both concerns in 1) figure 2D and 2) figure S3E. (Although it is unclear when the data was collected from reading the legend of Figure 2).

14. Figure 3A has a different pattern from that of Figure 2A. The authors explained the reasoning behind the change of pattern in the rebuttal comments and it is clear in the manuscript. This concern was addressed.

15. Legend of figure 4 was unclear and confusing as to what was in the figure. More detail was added to the legend. The authors addressed this concern.

16. Figure 4B was not referred to in the text and there was no description of how the single cell transcriptomics were done. The authors amended this, thus addressing the concern.

Reference cited:

Funkhouser-Jones LJ, Ravindran S, Sibley LD. 2020. Defining Stage-Specific Activity of Potent New Inhibitors of *Cryptosporidium parvum* Growth In Vitro. *mBio*. 11(2):e00052-20.
doi:10.1128/mBio.00052-20.

REVIEWER COMMENTS

Brief note: the authors response is in blue and italicized. DNRA stands for Does Not Require Action.

Reviewer #1 (Remarks to the Author):

thank you for thoughtfully addressing all of my comments.

Reviewer #3 (Remarks to the Author):

The clarity of the paper has been markedly improved by changing the title and by the addition of Fig. S2A. It is now clear that the cultures analyzed were 2-D hepatocyte cultures that were derived from organoid cultures. Also, the conclusions are strengthened by addition of the BLT1 dose-response and toxicity data; by showing the effect of a 2nd SRB1 inhibitor (ITX5061); and by correcting the cluster mislabeling. However, there are 2 remaining areas that could improve the paper:

1. Demonstrating that a 2nd SRB1 antagonist is active strengthens the finding that SRB1 has a role in Pf development. However, since up regulation of SRB1 expression after infection is a major finding presented in this paper, it still requires confirmation by a method other than scRNA-Seq. I am not sure why cell sorting cannot be done with analysis of infected cells; but could in situ hybridization or some other technique be used for analyzing for Pf and SRB1 co-expression. If this is too difficult, I understand.

To meet the reviewers concern, we have sourced an antibody against SRB1 [1, 2]. Specific SRB1 staining shows random presence in all cells [1] while Langlois et al only used this antibody for western blot and flow cytometry purposes. We performed staining on existing coverslips of PfNF175 infected Huheps. Infected cells tend to have more SRB1 staining compared to uninfected cells in the vicinity (See Figure S12 – two examples of infected host cell versus uninfected host cell). This data has been incorporated into lines 186 to 188. While figure 2I and J showed a significant upregulation in SRB1 transcripts in both cluster 3 (cluster with most infected cells) and for cells containing high parasite transcripts (regardless of the cluster) respectively, the effect is less obvious at a protein level when viewed using immunofluorescence. A possible hypothesis could be a higher turnover rate of SRB1 proteins in infected hepatocytes, although that would be technically very challenging to address in Pf liver stages currently.

2. Since the analyses use 2D-organ derived hepatocyte, why are primary hepatocytes used for toxicity testing instead of the organ derived hepatocytes? To be useful for drug screening, efficacy and toxicity should be tested in the same system with the same cells.

We thank the reviewer for this comment and have performed the toxicity test in KK2 for both BLT1 and ITX5061. The panel with the results is now shown in supplementary figure 7A. The method has been added on lines 441 to 444. The toxicity profiles of the treated KK2 with BLT1 and ITX5061 do not look overtly different from that of the primary human hepatocytes. Some

toxicity (> 20%) is seen at the higher concentrations of ITX5061 (80 and 100uM) which was to be expected. We have added the lines 189-192 to describe the results seen.

Intentionally Signed by Gary Peltz

Reviewer #4 (Remarks to the Author):

Reviewer #2

Overall, the authors addressed most minor concerns and some major concerns of this reviewer.

Have authors addressed major concerns of this reviewer?

1. This reviewer was concerned that the organoid model would yield significantly different inhibitory concentrations of atovaquone between the HuHep and HepOrg models based on previously published organiod-parasite systems (See Funkhouser-Jones et al 2020). The reviewer suggested that the authors include dose-response curves and EC50 of atovaquone in infected HuHeps and HepOrgs. The author's included a dose-response curve and EC50 of atovaquone only in HuHeps and not for HepOrgs (Fig.S1C). The authors did express that they observed a "significant decrease" of atovaquone at 10 nM in the HepOrgs that was not observed in the HuHeps (Fig.S1A). However, since it is unclear that whether this experiment was performed with three biological replicates and there is no statement regarding statistical analysis for these data, the claim of significant differences in the effect of atovaquone on parasite inhibition in both cell culture models remains unclear. Therefore, this concern was not addressed by the authors and requires further investigation.

We thank the reviewer for providing the reference. We have amended the legend of the figure in question by stating that the data is from one biological experiment with three technical replicates. This prevents us from making statistical conclusions and we apologise for our poor choice of wording in "significant decrease" which should have been "substantial decrease". We would have liked to address this point further however the more recently occurring inability of PfNF175 to generate gametocytes and therefore sporozoites interrupts our ability to conduct experiments as requested. However, we feel this does not detract from the claim that PfNF175 infected HepOrgs are susceptible to the anti-malarial compound atovaquone (regardless of the EC50).

2. This reviewer was concerned as to why the authors did not use the HepOrg model to explore the effects of the SR-B1 inhibitor: BLT1, and the role of SR-B1 in parasite development. The authors rebuttal expressed that they used HuHep because they are a "more biological relevant" cell line. Simply, the authors performed scRNAseq on HepOrgs, found differential expression of lipid metabolism related genes (including SR-B1) between infected and uninfected HepOrgs, and used HuHep for phenotypic studies to support their hypothesis of SR-B1's involvement for parasite development in hepatocytes. Based on this, it appears that while the authors developed a successful organoid model for parasite infection and transcriptomic analyses, the authors encountered technical limitations whereby phenotypic studies were unable to be conducted in the HepOrg system. There is no sentence

in the manuscript that explicitly justifies or explains the use of HuHeps for the phenotypic studies instead of the HepOrgs. Therefore, this concern was not addressed by the authors. However, the use of HuHeps to complement the scRNAseq findings provide useful data that sustains the authors' hypothesis.

We thank the reviewer for his/her understanding regarding our technical difficulties and have changed the result section to "HuHeps, the more directly biologically relevant cells, were treated with BLT-1 (20 μ M) at different time points (Fig. S8)..." (lines 194-195). Furthermore, in the discussion, we have included the sentence "Unfortunately, we were unable to test the impact of SR-B1 inhibitor on infected HepOrgs due to progressive loss in capacity of PfNF175 to transmit and generate sporozoites, [37, 38]" (lines 303-305).

3. This reviewer was concerned with the data visualization adopted in Figures 5 and 6. The authors chose to visualize the data using dots connected by lines, and this reviewer thought that it was a misrepresentation of the data. This reviewer suggested that the data should be represented in bar graphs depicting the means and their error bars. The authors followed this reviewer's suggestion and implemented bar graphs in these figures and included statistical analyses. However, the figure legends were not edited to reflect the changes and still give the impression that there are dots in the graph (i.e. Legend of Figure 5 says: "Each dot represents a biological experiment", while there are no dots in the graph.)

Addressed with the following sentences in Figures 5 and 6: "Each bar represents the average and error bars (SD) of three technical replicates from two biological experiments.... A total of two biological experiments were performed, each with two technical replicates: at least 100 schizonts were measured from each technical replicate except for NF54 and the BLT1 treated conditions as there are not that many schizonts surviving."

Have authors addressed minor concerns of this reviewer?

1. The overall concern was that all figures were referred to in the text, that there was enough detail in the figure legend to understand the experiment and that the figure labels are large enough to be read. Each figure is now referred to in the text, the figure legends adequately explain the figure (although more detail would be useful), and the figure labels are an appropriate size. This concern was addressed.

DNRA

2. There were concerns in lines 101 and 102. In line 101 the statement is that the size of schizonts is not significantly different between HuHeps and HepOrgs according to Fig1B. In the figure, there is an apparent difference in schizont size, and, although the authors state that there is no significant difference, no statistical test is mentioned to support this conclusion. In line 102, it was not clear that HepOrg A and HepOrg B were two different experiments. The authors corrected this and it is now explicitly stated. This concern was partially addressed.

We thank the reviewer for this point and has performed statistical analysis using the Welch's t-test on area under the curve. We have now updated the figure legend to say "Areas under the schizont growth curves of Huheps and respectively HepOrg A ($p=0.001$) and HepOrg B ($p=0.0002$) were significantly different (Welch t-test)."

3. Paragraph starting in line 118 lacked an explanation as to why the authors decided to use

the HepOrgs in 2D format to conduct the transcriptome analysis. The authors now have an explanation for this approach. This concern was addressed.

DNRA

4. Line 183-184 had a misleading word. The authors removed it. This concern was addressed.

DNRA

5. Line 186-188 was missing a reference. The authors included it. This concern was addressed.

DNRA

6. Line 195 had a misleading phrase and lacked statistical test to support the differences observed with BLT-1. The authors rephrased the line and included statistical tests. This concern was addressed.

DNRA

7. Lines 201-211 stated conclusions that did not directly translated from the experiments on MSP1 expression as a result of SRB1 inhibition. The authors softened the conclusions based on the reviewers comments. This concern was addressed.

DNRA

8. In lines 274-281 the authors referred to a “rodent model” to illustrate findings from past studies with Plasmodium species that infect murine organisms. The use of “rodent model” is confusing because it suggests the use of in vivo studies. This reviewer suggested to use the correct terms. While the authors address the concern of the misleading phrase in these lines, the use of “rodent model” can be still be read throughout the manuscript (e.g. Lines 204 and 282). Thus, the authors partially addressed this concern.

For line 209, it has now been modified to “*that P. yoelii and P. berghei (Plasmodium species of murine malaria models) must scavenge host lipids for their liver development*”. For lines 286 and 287, “*the liver stage development of P. berghei*” has been added.

9. The statement on lines 284-285 was missing a reference. The authors addressed this concerned and included a reference for the statement.

DNRA

10. Lines 294-299 stated conclusions that did not directly translated from the experiments on MSP1 expression as a result of SRB1 inhibition. The authors softened the conclusions based on the reviewers comments. This concern was addressed.

DNRA

11. Figure 1 labels were not consistent throughout the legend. The authors addressed this concern and the labels are consistent.

DNRA

12. Figure 2A was mislabeled. The authors addressed this concern and revised the labeling of the figure.

DNRA

13. In figure 2, it was unclear 1) what were the cutoffs of infection for the categorization of “low” and “high” parasite transcripts and 2) at what day post infection the data was collected. The authors addressed both concerns in 1) figure 2D and 2) figure S3E. (Although it is unclear when the data was collected from reading the legend of Figure 2).

We have addressed this concern by expressing the day of collection in the specific figure legend.

14. Figure 3A has a different pattern from that of Figure 2A. The authors explained the reasoning behind the change of pattern in the rebuttal comments and it is clear in the manuscript. This concern was addressed.

DNRA

15. Legend of figure 4 was unclear and confusing as to what was in the figure. More detailed was added to the legend. The authors addressed this concern.

DNRA

16. Figure 4B was not referred to in the text and there was no description of how the single cell transcriptomics were done. The authors amended this, thus addressing the concern.

DNRA

Reference cited:

Funkhouser-Jones LJ, Ravindran S, Sibley LD. 2020. Defining Stage-Specific Activity of Potent New Inhibitors of Cryptosporidium parvum Growth In Vitro. mBio. 11(2):e00052-20. doi:10.1128/mBio.00052-20.

1. *Ng, S., et al., Human iPSC-derived hepatocyte-like cells support Plasmodium liver-stage infection in vitro. Stem Cell Reports, 2015. 4(3): p. 348-59.*
2. *Langlois, A.C., et al., Molecular determinants of SR-B1-dependent Plasmodium sporozoite entry into hepatocytes. Sci Rep, 2020. 10(1): p. 13509.*

REVIEWERS' COMMENTS

Reviewer #3 (Remarks to the Author):

The authors have addressed my remaining concerns.

Intentionally signed Gary Peltz

Reviewer #4 (Remarks to the Author):

The authors have addressed all the concerns of this reviewer. Thank you for your work.